

# Mineralogy and mixing state of North African mineral dust by on-line single-particle mass spectrometry.

Nicholas A. Marsden[1], Romy Ullrich[2], Ottmar Möhler[2], Stine Eriksen Hammer[3], Konrad Kandler[3], Zhiqiang Cui[4], Paul I. Williams[1,5], Michael J. Flynn[1], Dantong Liu[1], James D. Allan[1,5], and Hugh Coe[1]

[1]School of Earth and Environmental Sciences, University of Manchester, UK
[2]Institute of Meteorology and Climate Research, Karlsruhe Institute of Technology, Germany
[3]Institute of Applied Geosciences, Technische Universitat Darmstadt, Germany
[4]School of Earth and Environment, University of Leeds, UK
[5]National Centre for Atmospheric Science, Manchester, UK

*Correspondence to:* Hugh Coe (hugh.coe@manchester.ac.uk)

**Abstract.** The mineralogy and mixing state of dust particles originating from the African continent influences climate and marine ecosystems in the North Atlantic due to its effect on radiation, cloud properties and biogeochemical cycling. Single-particle mineralogy and mixing state is particularly important in many processes but is difficult to predict because of large temporal and spatial variability and the lack of in-situ measurements of dust properties during emission, transport and deposition. This lack

of measurements is in part due to the remoteness of potential source areas (PSA) and transport pathways, but also because of the lack of an efficient method to report the mineralogy and mixing state of single particles with a time resolution comparable to atmospheric processes.

    In this work, the mineralogy and mixing state of the fine fraction ($< 2.5 \mu m$) in laboratory suspended dust from the Sahara and Sahel were made using novel techniques with on-line single-particle mass spectrometry (SPMS) and traditional off-line

scanning electron microscopy (SEM). A regional difference in mineralogy was detected, with material sourced from Morocco contained a high number fraction of illite like particles in contrast to Sahelian material which contains potassium and sodium depleted clay minerals like kaolinite. Applying the same methods to ambient measurement of transported dust in the marine boundary layer at Cabo Verde in the remote North Atlantic enabled the number fractions of illite/smectite clay mineral (ISCM), non-ISCM, and calcium containing particles to be reported at a 1 hour time resolution over a 20 day period alongside internal

mixing with nitrate, sulphate and organic/biological material. The ISCM and nitrate content was found to change significantly between distinct dust events, indicating a shift in source and transport pathways which may not be captured in off-line composition analysis or remote sensing techniques.

    The results show SPMS and SEM techniques are complimentary and demonstrate that SPMS can provide a meaningful high resolution measurement of single-particle mineralogy and mixing state in laboratory and ambient conditions. In most cases,

the mineralogy varies continuously between particles rather than a collection of discrete mineral phases. These techniques will be useful in resolving the complexity of mineral dust transport and in obtaining atmospherically relevant test material for laboratory experiments of dust properties.





*Copyright statement.* TEXT

# 1 Introduction

Aeolian dust derived from sources in North Africa has a substantial impact on the climate and ecosystems in the North Atlantic
but our ability to simulate climate response is poor (Evan et al., 2014; Kok et al., 2018). During aeolian transport, mineral dust

influences many atmospheric processes, many of which are a function of mineralogy as well as particle size and elemental
composition (Formenti et al., 2011). In addition, deposition of terrestrial material into the North Atlantic ocean is important to
the biogeochemical cycle of marine ecosystems as it represents a major source of iron, nitrogen and phosphorous (Baker et al.,
2003).

Major dust emission occurs in sparsely vegetated arid areas, but only when the surface properties and meteorological con-

ditions are favorable (Knippertz and Todd, 2012) which can be difficult to predict in dust emission models (Heinold et al.,
2013). During the summer months, the process of dust advection is associated with moist convection (Schepanski et al., 2009)
and results in a complex structure of stratified layers of warm dry air with very high dust concentrations (Dunion and Marron,
2008). Satellite observation show that these layers can be transported in discrete plumes thousands of miles westwards over
the Atlantic Ocean towards the Caribbean basin (Kaufman et al., 2005; Zhu et al., 2007; Doherty et al., 2008; Tsamalis et al.,

15 2013).

The source of these dust plumes is arid soils that consist of a complex mixture of organic and inorganic material. The
inorganic material largely consists of silicates (Quartz, feldspars, clay minerals), carbonate and metal oxides that occur as whole
grains or mixed mineral phases. The abundance of mineral types has a strong grain size dependence, with quartz occurring
in the coarse fraction and clay minerals dominating the fine fraction, but the size distribution is modified during emission

(Perlwitz et al., 2015a) so that the source soil may not be completely representative of the lofted mineral dust aerosol. Aircraft
observations have demonstrated that the size distribution of dust plumes evolve during transport which effects optical properties
(Ryder et al., 2013), but the contribution from associated changes in composition are not understood.

High temporal resolution measurements of dust mineralogy are needed to inform accurate simulations of dust emission,
transport and deposition processes in the present day and in the paleoclimate. For example, the provenance of mineral dust has

been determined by comparing mass fraction ratios of mineral types to the bulk analysis of soils in potential source areas (PSA)
(Caquineau et al., 2002; Scheuvens et al., 2013). Although improved soil maps are available for North Africa (Claquin et al.,
1999; Nickovic et al., 2012; Journet et al., 2014), they still lack the spatial resolution required to represent the heterogeneity
of soil within a PSA and results in an over simplification of the mineralogical relationships in a given soil type (Scanza et al.,
2015). Improvements to emission simulations require validation measurements of size resolved dust mineralogy and mixing

state at varying distances from the source (Perlwitz et al., 2015b).

Identification of the source of transported dust is problematic due to the inhomogeneity of source sediments and the mixing of
material during transport. Nevertheless, analysis of bulk mineralogical composition of source sediments can provide markers,
with illite/kaolinite ratios being particularly useful (Caquineau et al., 2002). At the Cabo Verde Islands, Glaccum and Prospero



(1980) analysed day long exposures of filters during the summer of 1974 and concluded that illite and mica were the dominant mineral types deposited in the tropical North Atlantic. In contrast, Kandler et al. (2011a) reported kaolinite as the dominant mineral type and attributed this to the dominance of sources in the Sahel during the summer.

In recent years, the composition of transported dust has attracted extra attention because mineralogy is thought to influence the ice nucleating (IN) ability of single particles (Zimmermann et al., 2008; Connolly et al., 2009; Möhler et al., 2008; Cziczo et al., 2009; Niedermeier et al., 2011; Augustin-Bauditz et al., 2016) and therefore influences climate by changing the radiative properties of ice and mixed phase clouds (Rosenfeld et al., 2001; DeMott, 2003), which represent some of the biggest uncertainties in weather and climate change prediction (DeMott et al., 2010; Carslaw et al., 2013). The mechanisms involved are not well understood and many analytical techniques have been developed to help further our understanding of this important process (Hoose and Möhler, 2012; Hiranuma et al., 2015). Recently, it has been demonstrated that mineral phase has an influence on heterogeneous freezing temperature, with some feldspar group minerals proving to be particularly efficient ice nucleating particles (INP) in droplet freezing experiments with nominally pure laboratory prepared mineral samples (Atkinson et al., 2013; Harrison et al., 2016; Peckhaus et al., 2016). However, the IN fraction in natural dust samples is much more difficult to assess due to complex mineralogy and mixing state that is difficult to resolve.

To complicate things further, cloud chamber studies of silicate mineral dust coated with secondary sulphate and organics have demonstrated that this mixing can increase hygroscopicity and lower the ice nucleation efficiency of a particle (Möhler et al., 2008; Niedermeier et al., 2011; Reitz et al., 2011). In addition, laboratory studies have suggested that the internal mixing state, particularly metal oxide - clay mineral assemblages, has important influence on radiative properties of dust (Nousiainen et al., 2009; Jeong and Nousiainen, 2014; Kemppinen et al., 2015; Di Biagio et al., 2017; Caponi et al., 2017). This highlights the need for measurement of single particle composition, including both mineralogy and internal mixing state in order to provide better constraints for global climate models (Kok et al., 2017).

An established approach to both ambient and laboratory measurements is to collect particles on filters and impactors for subsequent off-line analysis. Bulk mineralogy can be obtained by X-ray diffraction (XRD), and the composition of individual particles is assessed by advanced microscopy. Scanning Electron Microscopy (SEM) combined with energy-dispersive X-ray spectrometry (EDXS) is a favored method for elucidating composition because of the ability to probe the composition of individual particles. However, there are a number of drawbacks to this approach including; labor intensive post sampling analysis, difficulty in leveraging the full quantitative capability of SEM due to particle morphology effects, and loss of temporal resolution in dynamic situations. Consequently, small number of particles are reported in compositional groups, such as the frequently used scheme described by Kandler et al. (2009), which describes the dominant elemental features but not mineralogy.

A second, but less common approach is to use single-particle mass spectrometry (SPMS) to provide an on-line assessment of the composition. An advantage is that the high temporal resolution allows the evolution of the particle composition to be measured in a dynamic setting (Cziczo et al., 2006; Zelenyuk et al., 2009; Gaston et al., 2013), and the instruments can be deployed in-line with other techniques such as counterflow virtual impactor (CVI) to measure ice residuals (Cziczo et al., 2003; Gallavardin et al., 2008; Baustian et al., 2012; Zelenyuk et al., 2015; Worringen et al., 2015; Schmidt et al., 2016). However, compositional analysis with SPMS is fraught with difficulties relating to poor reproducibility, and the non-quantitative nature of





the measurement associated with instrument function and particle matrix effects (Reilly et al., 2000; Sullivan and Prather, 2005; Murphy, 2007; Hatch et al., 2014). These problems are particularly problematic for the analysis of irregular shaped particles composed of mineral phases whose elemental composition are similar. Consequently, SPMS has been used to discriminate mineral dust particles from other refractory aerosol types such as sea salt (Sullivan et al., 2007; Dall'Osto et al., 2010; Fitzgerald
et al., 2015; Schmidt et al., 2016), but not differentiate mineralogy.

In this study, a modified laser ablation aerosol particles time of flight (LAAPTOF) mass spectrometer (AeroMegt GmbH) was used to evaluate the composition of laboratory suspended soil, and ambient transported dust in the remote North Atlantic. A method for describing particle mineralogy and mixing state using sub-compositional analysis is introduced and demonstrated with nominally pure mineral samples. We applied this method to suspended natural soils sampled from two PSA in
North Africa, sampled at the Aerosol Interactions and Dynamics in the Atmosphere (AIDA) facility at the Karlsruhe Institute of Technology (KIT), as part of the ice nucleation research unit (INUIT) program studying heterogeneous ice formation in the atmosphere (https://www.ice-nuclei.de/the-inuit-project/). These results are used to interpret ground based ambient measurements of transported dust using a LAAPTOF at Cabo Verde during the summer of 2015 as part of the Ice in Clouds Experiment-Dust (ICE-D) campaign.

## 2   Methods

In order to characterize North African mineral dust aerosol, we compared laboratory suspended soil samples collected from PSAs in North Africa during the INUIT09 campaign, with ambient measurements of transported dust in the marine boundary layer (MBL) at Cabo Verde in the tropical North Atlantic during the ICE-D experiment. In order to make this comparison, we introduce reproducible methods of describing the mineralogy and mixing state of single dust particles extracted from sus-
pension using crystal structure and sub-compositional analysis with a LAAPTOF instrument. These techniques are described below, along with SEM EDXS methods used to obtain elemental composition of the laboratory suspended dust for method validation. The INUIT09 and ICE-D measurement experiments are also described in this method section.

### 2.1   LAAPTOF Single particle mass spectrometry

In single particle mass spectrometry (SPMS), aerosol is directly introduced into the instrument via a pumped inlet so that sample
collection and preparation are not required. Particle composition is analysed by time-of-flight mass spectrometry (TOFMS) after laser desorption ionisation (LDI) of individual particles using a high powered UV laser. The LAAPTOF instrument features an aerodynamic lens inlet, optical particle detection, LDI ion source and bipolar TOFMS as previously described by Gemayel et al. (2016). The instrument is capable of providing size resolved composition measurements in the size range approx. $0.4 - 2.5\mu m$. The instrument used in this investigation features a particle detection system based on a fibre-coupled
laser delivery that allows the simple interchange of laser systems of different wavelengths. Within the instrument, LDI was performed by ArF excimer laser (model EX5, GAM Laser Inc.), set to deliver 3-5mJ/pulse of 193nm radiation with a pulse width of 8ns.



**Table 1.** LAAPTOF setup parameters that differ between the laboratory and ambient acquired datasets

| Parameter | Laboratory | Ambient |
|---|---|---|
| Aerodynamic Sizing | no | yes |
| Detection laser wavelength | 488 | 532 |
| Excimer pules energy | 4mJ | 3mJ |
| Signal line attenuation | high | low |

*Signal line attenuation changes the saturation and limit of detection of the ion detection system

The SPMS technique is subject to a number of measurements biases with regard to particle size, shape and composition that result in preferential transmission, detection and ablation efficiencies of certain particles (Murphy, 2007). Table 1 documents differences in instrument setup between the laboratory and ambient measurements that may impact the comparison of the two datasets. For a detailed review of how these phenomena affect the LAAPTOF measurements, the reader is referred to Marsden
et al. (2016) and references therein.

In a previous study, Shen et al. (2017) showed that the LAAPTOF is capable of distinguishing the composition of different particles in the ambient environment using a combination of fuzzy clustering and reference based spectra classification, whilst Ahern et al. (2016) demonstrated a positive correlation between ion signals of organic markers and the quantity of condensed organic coating, despite inhomogeneity in the soot core of the particles. These studies show that the LAAPTOF is an appropriate
platform for studying the composition and internal mixing state of refractory aerosol.

### 2.1.1 Crystal structure analysis of silicates

Silicates are the most important group of rock-forming minerals and comprise a huge spectrum of compositions and crystal structures. However, the fine fraction of continental soil largely consists of clay minerals with minor amounts of feldspar and quartz (Perlwitz et al., 2015b). Clay minerals are phyllosilicates which consists of 2D layers of silicon tetrahedrally co-
ordinated with oxygen, and aluminum octohedrally co-ordinated with oxygen and/or hydroxyl. A simple example is that of the kaolinite, in which these two layer types are repeated to form a 3D structure referred to as 1:1 layer structure (Fig. 1(a)). A more complicated structure results in the presence of large cations of potassium, sodium and calcium. The charge and ionic radius if these cations is accommodated in a separate sheet of interstitial cations that must be charged balanced by the tetrahedral and octoherdal layers. In the case of potassium, charge balance is achieved with the common 2:1 layer structure of illite (Fig. 1(b))
in which there are two tetrahedral layers for each octoherdal layer.

In theory, these structures are free to polymerise uniformly in 2 dimensions. However, this is rarely realised because potassium, sodium and calcium are interchangeable as interstitial cations so the composition is not necessary uniform throughout the crystal lattice. Indeed, the illite structure in Fig. 1(b) is the idealised version of the pure mineral, which is often referred to as an end-member of the illite series. Complete uniformity of clay mineral composition is unlikely to exist and on the scale of
a single particle or between particles from the same soil sample.





Larger and more uniform crystals of feldspars are more common due to their formation from molten material rather than through the alteration of other minerals as is the case for clay minerals. In K-feldspar, such as microcline and orthoclase, interstitial cations of potassium are fixed in a cavity where they are charge balanced by a 3D network of tetrahedrally co-ordinated silicon and aluminum (Fig. 1(c)). Substitution of potassium, sodium and calcium is not readily achieved in the solid

phase like the clay minerals, but dissolution of these cations results in alteration of feldspar to clay minerals during diagenesis of sediments. For example, K-feldspar can be progressively altered to form quartz and kaolinite if the potassium is removed by dissolution.

In a study of ion formation mechanism, Marsden et al. (2018) showed that the ionisation potential and co-ordination of interstitial cations has a strong influence on how the structure breaks apart during LDI process of silicate particles. This is

exploited for the differentiation of mineral phase by the measurement of average ion arrival of the $O^-$ and $SiO_3^-$ ion species, which is thought to record the initial kinetic energy of the ionic fragments during ablation.

Expressed as a peak shift ratio ($\tau$), which can be compared to nominally pure mineral samples on a particle by particle basis, laboratory calibrations have demonstrated that for illite-smectite clay minerals (ISCM) the abundance of exchangeable sodium and potassium cations results in a unique range of $\tau$ values between 0.2-0.58. Feldspars typically produce $\tau$ values >0.58 (Fig.

2) due to the interlocking nature of the interstitial cation in their crystal structure. Kaolinite and amorphous glass produces $\tau$ values close to 1 due to the complete absence of interstitial cations and lack of crystal structure respectively.

### 2.1.2 Sub-compositional analysis of silicates

Sub-compositional analysis is used to produce relative composition measurements that can be compared to fingerprints generated from nominally pure mineral samples. Silicate composition is considered with the ternary system $Al^+ + Si^+, K^+, Na^+$

which represents the cations readily observed in the SPMS of mineral dust (m/z 27, 28, 39, 23 respectively). The $K^+$ signal is a useful reference as its ionization energy is the lowest of the common cation signals, meaning it is less likely to be perturbed by the presence of other elements. This approach differs from the $Ca^+, Fe^+, Al^+$ ternary system used by Sullivan et al. (2007) which could be significantly perturbed by variations in $K^+$ and $Na^+$ due to the matrix effect. In addition, $Ca^+$ is not considered due to problems differentiating this signal from the potassium signal at $m/z 39 - 40$, and the potential for interference

from calcium within carbonate minerals.

Source clay material from the clay mineral society and crushed feldspar crystals were suspended in a home made dust tower in the experiment described in Marsden et al. (2018). The samples were representative of the end members of the clay minerals and feldspars with varying alkali metal content. Elemental composition and mineralogy of these samples have been previously characterised and can therefore be used as reference fingerprints for comparison with uncharacterised natural soil. Ternary

diagrams of the clay minerals and feldspar standards are displayed in Fig 3.

Although the measurement is clearly non quantitative with respect to bulk XRF analysis, the relative measurement produces a clear separation between the clay minerals, and between K and Na rich feldspars. The 2D space in these diagrams are also non-linear due to the matrix effect, so that quantitative composition cannot be deduced from these plots alone. Consequently, clay minerals and feldspars are not easily distinguishable from each other despite differences in actual composition, but it is





interesting to note that K-feldspar appears less K and Na rich than illite clay despite the structural formula and XRF analysis of the bulk sample indicating the contrary. This is because the tetrahedral framework of the feldspar makes the release of interstitial $K^+$ and $Na^+$ without also releasing $Al^+$ improbable, whereas they can be released independently of $Al^+$ from the weakly bonded interlayer in 2:1 clay structure.

### 2.1.3 Sub-composition analysis of internal mixing state

Internal mixing of non-mineral species can occur during soil formation or during transport in the atmosphere where heterogeneous reactions with ozone take place on the surface of the particle (Usher et al., 2003). Reactions with nitric, sulphuric and organic acids can produce nitrate, sulphate and organics on the particle surface respectively. Reference spectra obtained with the LAAPTOF during the FIN-1 campaign show the presence of sulphate marker $HSO_4(m/z\,97)$ on particles after mixing suspended feldspar with ozone and sulphuric acid in the AIDA chamber (Fig. 4(b)). Similarly, organic markers $C_2, C_2H$ and $C_2H_2(m/z\,24, 25, 26)$ appear on particles after mixing suspended feldspar with ozone and $\alpha$-pinene (Fig. 4(c)). These ion combinations have also been observed on ambient mineral dust using SPMS instruments (Silva and Prather, 2000; Sullivan et al., 2007; Fitzgerald et al., 2015).

In ambient dust, organic or biological material may have been mixed with a dust particle in the soil before emission. Indeed, the transport of microorganisms in dust storms is human health concern (Griffin, 2007). Recently, Yamaguchi et al. (2012) provided direct evidence of bacterial cells on Asian dust particles and demonstrated a global dispersion pathway through dust transport. The internal mixing of biological material with dust particles is therefore of interest. LAAPTOF reference spectra of bacteria, also from the FIN-1 campaign, show strong signals of $CN$ and $CNO(m/z\,26, 42)$ (Fig. 4(d)) representing fragments of nitrogen containing organic compounds. These markers have been attributed to compounds of biological origin (Pratt et al., 2009; Cahill et al., 2012; Creamean et al., 2013), although other studies have found these markers in aerosol where biological components were not expected (Sodeman et al., 2005; Zawadowicz et al., 2016; Wonaschuetz et al., 2017). Consequently, we describe these markers as organic-biological (Org-Bio hereafter).

Internal mixing of natural soil samples with non-minerals in this work are considered with the ternary sub-composition $Cl^-, CN^- + CNO^-, SO_4^-$. These markers represent compounds that could be present in soils and ambient dust, but are not present in silicate minerals structure. The inclusion of the $Cl^-$ is important because it is a non-silicate element with a very high electron affinity and is therefore included as a reliable reference.

## 2.2 Scanning electron microscopy

Particles were sampled from the AIDA chamber with a single-stage nozzle impactor (50% low cut-off diameter at approximately $0.1\mu m$ aerodynamic diameter; for specifications see Kandler et al. (2007)) on nickel grids (transition electron microscopy (TEM) grids type S162N9, Plano, Wetzlar, Germany). The inlet was connected to a sampling line on the AIDA chamber and the sample time varied between 12-16 min.

All samples were analysed with computer controlled scanning electron microscopy (ccSEM) using a FEI Quanta 400 FEG instrument (FEI, Eindhoven, The Netherlands) equipped with an energy-dispersive X-ray detector, X-Max150 (Oxford, Ox-





fordshire, United Kingdom), and the Oxford software Aztec (version 3.3 SP1). An acceleration voltage of 12.5 keV, spot size 5, a working distance of 10mm and high vacuum conditions ( 10-5 hPa) were used for all samples. Backscatter electron images were used to segment particles from substrate. The particles of interest were measured with 4 seconds counting time for X-ray microanalysis. The TEM grids were mounted in a copper sample holder (avoiding interference with chemical composition of the particle) equipped with a beam trap (maximizing the characteristic-peak-to-background ratio).

Chemical composition of all elements with atomic number higher than 3 (Li) was determined with energy-dispersive X-ray microanalysis. $ZAF$ correction (provided by the software) was used to correct for matrix effects (atomic number effect, absorption effect and the fluorescent excitation effect). A sorting step after ccSEM was performed to remove particles with low X-ray counts (due to shading effects) and features of the TEM grid. Between 159 and 776 alumosilicates were detected per samples with the highest abundance of particles at an average diameter of $200nm$.

The traditional method for reporting the composition of aerosol particles involves the classification of particles into compositional groups using a variety of elemental ratios and boundary rules e.g Kandler et al. (2007, 2011b); Young et al. (2016). However, these classification systems are impossible to apply to SPMS due the matrix effects that skews and suppresses elemental ratios. We therefore choose to display the SEM composition as ratios of Al/Si, (K + Na)/Si and (Fe + Mg)/Si ratios, which leverages the quantitative ability to produce a representation of the interstitial cation and aluminosilicate structure that can be intuitively compared to the ternary diagrams obtained by SPMS.

The average chemical composition of nominally pure mineral samples is plotted in Fig 6(a). This demonstrates that pure mineral can potentially be differentiated using quantitative $Al/Si$ ratio and the $(K + Na)/Si$ (cation to silicate) ratio. For example, the pure Kaolinite clay mineral has the highest $Al/Si$ ratios ($\approx 1$) but no $(K+Na)$ content. In general, the decreasing $Al/Si$ ratio must be balanced by increasing cation/Si ratio in clay minerals.

## 2.3 INUIT09 Laboratory Suspended Dust Experiment

One of the objectives of the INUIT09 campaign (July 2017) was to determined if single particle mass spectrometry is capable of differentiating between aerosol particles of different mineralogical compositions.

### 2.3.1 Experiment Setup

For the aerosol generation during the INUIT09 campaign, the natural dust samples were sieved to diameters less than $75\mu m$ and injected in either the AIDA cloud chamber or the Aerosol Preparation Chamber (APC). For injection, a rotating brush generator linked to a cyclone impactor was used, resulting in aerosol particles with aerodynamic diameters of less than about $5\mu m$. The dust number concentration in the AIDA chamber was initially between 200 and $300cm^{-3}$ and in the APC between 1000 and $5000cm^{-3}$. The number concentrations and size distribution of the dispersed aerosol are detailed in Supplement S1.

During this study, the single particle mass spectrometer LAAPTOF was either connected to the AIDA chamber or the APC, and sampled for about 1 hour during each experiment. During the sampling time, both the AIDA chamber and APC were hold to nearly constant temperature (either $-16\deg C$ or $-21\deg C$ in AIDA and $25-27\deg C$ in APC) and pressure ($998-1010hPa$ in AIDA and $990-1005hPa$ in APC) conditions.





### 2.3.2  Soil Sampling Locations

Soil samples from two specific ecoclimatic zones were suspended at the AIDA facility; the mountainous north-west margin of the Sahara Desert, and the west-central Sahel beyond the southern margin of the Sahara Desert (here-after named Sahel Dust). These PSA were chosen because there is likely to be differences in mineralogical composition between the two zones.

Analysis of surface material across North Africa are scarce, but the available studies demonstrate a North-South decease in illite/kaolinite (I/K) ratio in the clay fraction (Caquineau et al., 2002; Scheuvens et al., 2013; Formenti et al., 2014) mainly due to differences in diagenetic history between dry and wet climate zones.

The sampling locations of the natural arid soils are displayed in Fig. 5. Moroccan dust samples were collected in the Mhamid region, on the upper Draa valley, Morocco during the Saharan mineral dust experiment (SAMUM) (Ansmann et al., 2011). The

Draa valley is a large geographical feature on the Southern edge of the Anti-Atlas mountains, beyond which is the Sahara desert of Algeria. All sampling locations fall within unit 1336 in Digital Soil Map of the World (DSMW), compiled by the Food and Agriculture Organization (FAO) of the United Nations (FAO, 1995, 2007), which describes the dominant soil as fluvisols (50%), associated with yermosols (20%), regosols (20%) and solonchaks (10%).

- **DDS01** 29.83773 °N, -5.76143 °E River sediments

- **DDS02** 29.84957 °N, -6.01508 °E Hamada with large amounts of dust

- **DDS03** 29.86202 °N, -6.156760°E Border of dry salt/ silt plain of Lac Iriqui close to sand dune fields

The samples of Sahel soil were collected from diverse geographical locations in Niger and Burkina Faso. All three sampling locations fall within the zone of kaolinite rich soil (Fig. 5) where the DSMW indicates CEC <35 cmol/kg.

- **SDN02**

13.516667 °N, 2.633333 °E Banizoumbou, Niger

- **SDN05** 13.522203 °N, 2.133011 °E Grand Mosque in Niamey, Niger

- **DDA01** Dano, Burkina Faso

### 2.4  Ambient Measurement at Cabo Verde during ICE-D

The Cabo Verde archipelago is a favorable location for remote marine measurements. Situated some 800km off the West coast

of North Africa (Fig, 5), it is a site of long term measurements of greenhouse gasses, trace gasses and aerosol properties are recorded (Carpenter et al., 2010). Notable studies of mineral dust composition in the area include long term filter sample collection at Cape Verde atmospheric observatory (CVAO) (Patey et al., 2015) and Praia City, Santiago Island (Salvador et al., 2016), and filter collection at Praia Airport, Santiago Island for the SAMUM campaign January-February 2008 (Kandler et al., 2011a).





Ground based ambient measurements during ICE-D took place in the mobile Manchester Aerosol Laboratory located within the perimeter of Praia International Airport, Santiago Island, Cabo Verde, (14°57'N 23°29' W, 100 m asl), approximately 1500m from the coast and 150m from the airport runway. The main airport terminal and the outskirts of the city of Praia were 400m and 2500m downwind of the prevailing NE wind respectively. Aerosol was sampled via a 6 inch plastic inlet fixed to a

10m tower and pumped at 185l/min. The flow was distributed to a suite of on-line instruments, including the LAAPTOF, via a system of 48m ID lines isokinetically sampled from the main inlet, and heated to 19°C.

## 3  Results

In the analysis mineral dust, we choose to analysis the silicate mineralogy and the internally mixed non-silicate components separately using the methods described above.

### 3.1  Single particle analysis of laboratory suspended natural soil (INUIT09)

The sampling of particles took place after dispersing the source material in the APC after cleaning, and it was therefore expected that all particles were particles from the original soil. Initial examination of the mass spectra showed that the vast majority of particles contained the markers for silicates, indicating that all particles which produced a mass spectrum contained at least some silicate minerals.

### 3.1.1  Silicate mineralogy of laboratory suspended soil (INUIT09)

SEM EDX analysis of the suspended soil shows that the composition of single particles within each soil type varies as in a continuous distribution rather than in distinct clusters (Fig. 6). A distinct difference in the elemental composition of soil from the Northern Sahara and the Sahel exists in these diagrams, particularly with respect to the Al/Si ratio. The Moroccan soils (Fig. 6(b-c)) have a spread in compositions close to the illite plotted on the diagram, with the DDS01 samples being more

diverse than the DDS02. In contrast, the soil from the Sahel (Fig6(c-f)) consists of material with higher Al/Si ratio which approaches the composition of the plotted kaolinite. This is in agreement with cation exchange capacity (CEC) measurements of sediments in potential source areas (PSA) which show that kaolinite fraction in soils increases in soils in the Southern Sahel regions compared to the northern regions of the Sahara (Scheuvens et al., 2013).

SPMS mineralogical analysis proceeds by considering the relative abundance between $K$, $Na$ and $Al + Si$, which together

define the phase of many minerals in the continental crust. This sub-compositional analysis was previously applied to nominally pure mineral samples from the clay mineral society (CMS) to provide a fingerprint reference for the common clay mineral phases (Fig. 7(a)). The same analysis applied to Moroccan dust (Fig. 7(a-c)) and Sahelian dust (Fig. 7(e-g)) shows key differences between the composition of the two ecoclimatic zones, with Morrocan dust tending towards an illite composition and Sahelian dust tending to a kaolinite composition, in agreement with the SEM EDX analysis.

Application of the crystal structure analysis (color function in Fig. 7) to each particle indicates that the majority of the particles are of a 2:1 clay structure, even in Sahelian dust particles that have a kaolinite like composition. This may suggest



an impure kaolinite or montmorillonite that is not captured in the cation sub-composition due to the relatively high sensitivity to $K^+$ in our measurement. A comparison of the Al+Si sub-composition with respect to alkali metals obtained by LAAPTOF and SEM measurement (Fig. 8) demonstrates a much greater sensitivity to alkali metals than the established filter technique. Both techniques show a qualitatively higher Al+Si content in the Sahel compared to the Moroccan sample, but this is greatly

exaggerated in the SPMS analysis.

In general, our analysis suggests the fine fraction of these soil samples to consist of $K$ rich and $Al$ rich clay minerals. This agrees somewhat with a comprehensive study of soil samples using many techniques by Engelbrecht et al. (2016) that concluded suspended mineral dust particle were primarily made up of two types of mineral assemblages; i) fragments of micas, clays, oxides, and ions of potassium ($K^+$), calcium ($Ca^+$), and sodium ($Na^+$) rich colliods in amorphous clay-like material,

and ii) kaolinite with individual oxide mineral grains. The exception is that our measurements indicate that the potassium and sodium are retained in a 2:1 clay structure rather than in amorphous clay.

In order to explore the amorphous and felsic composition, we extracted particles with $\tau$ values of 0.58 - 0.8 into separate plots (Fig. 9). Perhaps unsurprisingly, the composition of these types of particles reflects the composition of clay particles (Fig. 7), with samples from the Sahel showing a greater loss of alkali metals due to weathering and diagenesis. It is not possible to

separate felsic and amorphous material with any confidence, but it is reasonable to say that any feldspars that may be present in the Moroccan samples are fresher (un-altered) than their counterparts from the Sahel.

The sample that has potentially the most feldspar grains is DDS01 which shows a small number of particles with a felsic composition in the SEM EDX analysis (Fig. 6(b)), and maybe a reflection of the river sediment containing freshly eroded rock from the Anti-Atlas. This sample also displays a distinct mode in the SPMS $\tau$ histogram in the felsic/amorphous region (Fig.

7(d)). In general, the SPMS crystal structure analysis in the $\tau$ histograms agree with the SEM EDX analysis of the distribution of composition in the samples. For example, the $\tau$ histogram of DDS02 indicates a predominately illite composition which is also reported in the narrow composition distribution in the SEM EDX analysis.

### 3.1.2   Internal mixing state of suspended natural soil

We approached the SPMS mixing state analysis using a similar method described by Sullivan et al. (2007) where the relative

signal of the chlorine ($Cl^-$) anion, ammonium ($NH_4^+$), and sulphate markers ($SO_4^-$) were considered, except we use the Org-Bio markers ($CN^-$ and $CNO^-$) instead of ammonium to avoid comparing positive and negative ion peak areas. In addition to being a potentially important compound in changing the physiochemical properties of dust particles, chlorine has a high electron affinity and therefore $Cl^-$ is a good reference signal, similar to the role of $K^+$ in positive ion mode.

Unlike the mineralogical composition, the mixing state of the natural soil samples varies quite a lot within the same PSA

(Fig. 10). All samples contain particles with significant amounts of chlorine. Sample DDS03, collected close to a salt/silt plain, shows a relatively high sulfate content, probably associated with evaporite deposits.

The biggest difference however is the relative amount of org-bio internally mixed in the particles. This is further demonstrated in the histograms of the normalised org-bio content displayed in Fig. 11. Most samples contain particles with sig-





nificantly less org-bio than chlorine, but samples DDS01 and SDN02, collected from a dry river bed and agricultural land respectively, have a number of particles where the org-bio signal is even stronger than that of chlorine.

## 3.2 The composition and transport history of transported dust (ICE-D)

In order to examine the properties of the transported dust in the ambient measurement, the mass spectra of silicate particles
must be extracted from the general aerosol population. This was achieved using the fuzzy c-means clustering supplied with the data analysis software (LAAPTOF Data Analysis v1.0.2). The analysis was performed on positive and negative ion spectra and on negative ion spectra only. In both cases, silicate and calcium classes of particles were reported along with sulphate, carbonaceous and abundant sea-salt particles. Analysis using only negative ion spectra proved more useful in the evaluation of mixing state because it was not subject to false class divisions because of peaks shifting in positive ion spectra. A full
description of the clustering analysis, including mass spectra of the cluster centers, is provided in supplement S2 and S3.

Averaged detection rate of the silicate and calcium particle types, as determined by the fuzzy clustering analysis, is displayed in Fig.12(a) with a 1 hour resolution . A previous comparison with an aerodynamic particle sizer indicated that the silicate particle count represented approximately $1\%$ of the actual silicate present due to instrument function (Marsden et al., 2016). Nevertheless, the temporal evolution of the particle classes is representative and compares well to the the dust fraction of
incandescent particles identified in tandem Soot photometer (SP2) measurement (Liu et al., 2018).

### 3.2.1 Calcium rich particles

Calcium carbonate ($CaCO_3$) is the most abundant non-silicate material in arid soils and is expected to be a major constituent in mineral dust aerosol. However, pure calcium carbonate particles were not observed in this ambient measurement. Alkaline calcium carbonate particles are expected to readily react with acid gasses in atmosphere to form calcium salts. The reaction of
calcium carbonate with nitric acid to form hygroscopic $Ca(NO_3)_2$ has been offered as a mechanism to explain deliquesced calcium particles on filters (Krueger et al., 2004; Laskin et al., 2005; Matsuki et al., 2005) collected after transport through urban and industrial areas. More recently, it has been suggested that in remote marine environments, a prevalence of $HCl$ over $HNO_3$ would favor the formation of $CaCl_2$ as the principal calcium salt (Tobo et al., 2010; Kim and Park, 2012).

A distinct calcium chloride particle class is reported in this ambient measurement. These particles are characterised by
signals of $Ca^+(m/z\,40)$, $CaO^+(m/z\,56)$ and $CaCl^+(m/z\,75,77)$ in positive ion spectra and $Cl^-(m/z-35,-37)$, and $CaCl_3^-(m/z\,145,147,149)$ in negative ion spectra (See supplement S3.2). The temporal evolution of this particle class is similar to that of the silicate particle class (Fig. 12(a)), but very different to that of the sea-spray particles (see supplement S3.5), therefore we conclude that is has a similar source to the silicate and probably represents calcium carbonate that has been processed in the atmosphere .

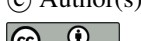



### 3.2.2 Temporal evolution of the silicate particle mineralogy

In a previous study using TEM of sliced dust particles collected in Tenerife, the majority of particles of transported Saharan dust were clay rich agglomerates composed of a ISCM matrix with inclusion of kaolinite and hematite (Jeong et al., 2016). In light of this, the silicate class of particles in our ambient study is analysed making the assumption that each mass spectra in the
ISCM category are representative of the average clay matrix in single particles.

The silicate class of particles is differentiated into ISCM and non-ISCM by crystal structure analysis to produce an time series of ISCM ratio (Fig. 12(b)). The ISCM ratio is the number of particles which have a $\tau < 0.58$ divided by the number of silicate particles which have $\tau > 0.58$, representing ISCM and non-ISCM (feldspar, amorphous and very pure kaolinite) respectively. The start of the measurement period is characterized by relatively high ISCM ratio which is generally »1. These
conditions persist up until 17th August when the ISCM ratio is much lower than in the previous period with (ISCM ratio <1).

The ISCM ratio and the average detection rate of silicate and calcium containing particles are combined to produce an hourly average of the ISCM, non-ISCM and calcium number fractions in mineral dust (Fig. 12(c)). It is noted that these fractions are relative to the detection efficiency of the instrument to each particle type, but the temporal evolution is representative. This display shows that an increase in calcium containing particles also occurs alongside the decrease in ISCM ratio on 17th August.
We use the time series to define two dust events; D1 centered on 00:00 11/08/2015 and D2 centered on 15:00 17/08/2018. Sub-compositional analysis of the two dust periods, using the same method as re-suspended dust, is displayed in Fig. 14. Dust event D1 has a composition that is relatively rich in illite with 2:1 phyllosilicate structure (Fig. 14(a)), similar to that observed in Moroccan soil DDS01 and DDS02 (Fig. 7(b, c)). This illite fraction is much reduced in dust event D2 which has a higher proportion of non-ISCM felsic/amorphous material (Fig. 14(e)).

### 3.3 Temporal evolution of the mixing state of silicate particles

The internal mixing of silicate dust is represented by the sub-composition of markers of non-silicates in similar manner to the treatment of suspended dust in section 3.1.2 except we consider nitrate in ambient particles instead instead of sulphate. Increases in nitrate have previously been observed in association with air masses passing through Europe and Africa to Cabo Verde (Fomba et al., 2014).

Nitrate mixing with the silicate dust is varied, with the sub-composition of nitrate markers indicating a relatively greater quantity of nitrate mixing with silicate particles during the inter-dust period between the defined dust events. This shows a similar trend to the number concentration of nitrate sea-spray aerosol particle types extracted by fuzzy clustering (Supplement S3.5), which suggest the particles may have co-existed in the same airmass for some period of time. There was a distinct drop in nitrate sub-composition during the dust period D2 when the crystal structure analysis suggests a change in silicate mineralogy.

Internal mixing with organic-biological material was much less varied than in the laboratory generated reference material. The number of particles that contain a significant fraction of this sub-composition (Org-Bio > 0.2) varies with the dust concentration (Fig. 12(d)), and the median amount of organic-biological marker signal is fairly stable, with the exception of a 24 hour period following dust event D1 (Fig. 12(e)).

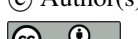



### 3.4 Potential source areas and transport history

Dust transport to Cabo Verde is controlled by the position of the Azores high pressure system whose anticyclonic flow results in persistent north easterly trade winds along the coast of Morocco, Western Sahara and Mauritania (Carpenter et al., 2010). Dust concentrations at ground level are influenced by the dust deposition rate which is strongly dependent on the horizontal and vertical dust distribution (Schepanski et al., 2008). During the summer months, the westward moving SAL typically occurs at relatively high altitudes (1-6km) in the Cabo Verde area (Tsamalis et al., 2013), and the warm, dry, dust-laden air creates a stable layer that is separated from the lower levels by a strong temperature inversion (Wong et al., 2009).

Back trajectory analysis of the dust events D1 and D2 shows transport in the trade winds within the marine boundary (Fig. 14c, g). The influence of air mass from the African continent during D1 and the high ISCM ratios for in these silicates ($68\%$) suggests this material originated from the Northern region of the Sahara. The back trajectories indicate that some of this air mass was lofted over the Atlas Mountains before descending into the MBL. Dust mobilisation associated with density currents related to moist convection has been observed on the Sahara side of the Atlas Mountain chain (Knippertz et al., 2007). Satellite observations show a plume of dust developing in Algeria and crossing the Atlas mountains and then above the Atlantic Ocean in the days proceeding the dust event D1 (Supplement S5). However, source activation observed by remote satellite is subject to errors associated with the temporal resolution of the measurement (Schepanski et al., 2012).

A feature of these back trajectories is that the air mass arriving during D2 came into direct contact with the coastline of Mauritania which is a possible source of dust. This could explain the relatively low ISCM content and relatively high calcium carbonate fraction which is consistent with source sediments of the Atlantic Coastal basin (Moreno et al., 2006). The particle size distribution of the silicate from D2 also suggests that this dust may have been transported a relatively short distance compared to D1 (Fig. 13). The silicate from D1 has a well-defined mode at approximately $1\mu m$, which is in contrast to the broader distribution of D2 particle sizes, which extends above $2\mu m$.

The mixing state of particles also suggests different transport pathways for the two dust events. The mixing of nitrate with silicate during and after D1 indicates heterogeneous reaction with nitric acid in polluted air and is consistent with transport from the North, whilst the cleaner particles in D2 indicate that no such mixing took place. An increase in the fraction of sulphate sea-spray aerosol in D2 on the other hand (Supplement S3.5) may be associated with biogenic sources from coastal upwelling off the coast of Mauritania.

### 4 Discussion

In this analysis, we assert that an approximation of the silicate mineralogy of the fine fraction can be made by considering the dust particles as clay mineral matrices with additional felsic/amorphous grains. We focus on the relative abundance the K, Na and the combined Al and Si signals, all of which are readily observed in the mass spectra. Although a non-quantitative measurement, this provides a relative assessment of the interstitial cation content relative to the tetrahedral and octohedral co-ordinated cations, which together define mineral phase. This makes the assumption that Mg, Fe and Ca are of minor importance




in silicate mineralogy in continental sediment. While certainly untrue at a local level, it is informative of the genesis and subsequent weathering history of the clay fraction in soil on a continental scale.

The ability of SPMS to report crystal structure enables the differentiation of clay minerals from felsic/amorphous material, which can then be analysed separately using sub-composition analysis. Although non-quantitative, the sensitivity of the SPMS technique to potassium and sodium makes it particularly suitable for differentiating kaolinite from illite in the fine fraction, but is less effective than SEM EDX at identifying silicate particles rich in magnesium and calcium due to suppression by the matrix effect and interference from poorly resolved peaks. However, the division of clay species is also difficult in off-line analysis and kaolinite and illite are often the only clays species detected by XRD in aircraft studies (Formenti et al., 2014), therefore these types of on-line measurements are likely to be of some use in the study of mineral dust.

Sub-compositional analysis is a techniques that is commonly used for the analysis of complex chemical relationships in circumstances in which the full composition of a substance is impossible to obtain, such as in whole-rock geochemical analysis, where it is convenient to describe the composition in terms of a relative measurement of carefully chosen components. The ternary diagram provides an intuitive way of displaying relative information whilst avoiding some of the sub-compositional incoherence associated with two component systems. With due respect to the matrix effect in SPMS, these diagrams can be used to provide mineralogy and mixing state of natural dust relative to laboratory generated standards.

Using sub-compositional analysis, a distinct illite rich composition is observed in the Moroccan samples compared to a kaolinite rich composition in samples from the Sahel region. The crystal structure analysis confirms a 2:1 phyllosilicate structure for particles with illite composition but also suggests a 2:1 structure in many particles with a Kaolinite composition, which may represent an impure kaolinite compared to the CMS standard (KGa-1b). Minerals undergo significant alteration and weathering in the soil making pure minerals difficult to identify, even by XRD of the bulk sample (Formenti et al., 2014). Nevertheless, the fraction of illite and non-illite particles is a useful indicator of PSA, similar to the illite/kaolinite ratio. Many more SPMS measurements are required to fully understand the clay ratio signatures at source and how this is maintained after varying distances of transport and mixing.

It was suggested by Scheuvens et al. (2013) that illite/kaolinite ratio >2 are indicative of the PSA that comprises the northern part of the west coast of Africa and the south eastern slopes of the Atlas Mountains. The Anti-atlas mountains was also suggested as a source of micronutrients to the North East Atlantic Ocean by Chavagnac et al. (2007). In off-line filters collected at Praia Cabo Verde, Kandler et al. (2011b) reported illite/kaolinite ratio of between 1:2 and 1:4 during winter transport of dust to Cabo Verde from coastal Mauritania and Mali respectively. In continuous long term measurements of elemental composition, Patey et al. (2015) reported seasonal variation in Al content which was attributed to seasonal changes in dust transport, and reported illite as the most abundant. In this later case, only a limited number of filter samples were selected for XRD analysis and the reported clay mineral fractions were subject to large errors due to low sample loading and preferred orientation of the grains. In all the above off-line studies, filters were exposed for around 1 day resulting in poor temporal resolution of the data. Our high resolution measurements show that the mineralogy of dust can change on a hourly basis, suggesting that changes in dust source and particle composition may be lost in off-line data.



The mechanism by which dust enters the marine boundary layer (MBL) is crucial to understanding its source and transport history in ground based measurements. Dry deposition of mineral dust from a lofted Saharan air layer (SAL) by gravitational settling is expected to be a slow process for particles of this size, although dust entrainment by convective processes may bring larger quantities of material from the free troposphere (Bravo-Aranda et al., 2015). The deposition rate of particles is an active topic of research with recent work suggesting a coarse mode is maintained longer than expected (several days) by atmospheric dynamics ((Denjean et al., 2015; Weinzierl et al., 2017)).

In a XRD analysis of material captured in marine sediment traps of Cape Blanc, Mauritania, Friese et al. (2017) concluded that seasonal variation in mineralogy occurred due to changes in transport pathways; from long-distance transport in the SAL during summertime, to local transport in the trade winds in winter. Our measurements also suggest two pathways for dust arriving at Cabo Verde, but with a much finer temporal resolution. We propose that in August 2015, illite rich dust from the NW margin of the Sahara was advected into the free troposphere and became mixed in to the MBL off the coast of Morocco before several days transport to Cabo Verde. An eastward shift in the Azores high results in the direct entrainment of dust into the MBL from the coast of Mauritania, producing a relatively illite poor and calcium rich dust event in Cabo Verde on 17th August that lasted for several hours. This is supported by the nitrate content which is high in the dust transported from the North of Cabo Verde, as demonstrated by Salvador et al. (2016).

One of the more surprising results of this work is the relatively small variation in the organic-biological mixing in comparison to the laboratory generated reference material. This is in spite of differences in source and transported history apparent in the mineralogical analysis. This may be due to a greater chlorine content that perturbs our relative measurements, but it is interesting to note that Price et al. (2018) did not see significant variation in ice nucleating particle (INP) concentration in the accompanying aircraft sorties despite geographically widespread dust sources. This is important because the efficient ice nucleating properties of proteinaceous INP can be transfered to dust particles after mixing ((Augustin-Bauditz et al., 2016; O'Sullivan et al., 2016)). The origin of the organic-biological markers $CN^-$ and $CNO^-$ in these dust particle requires further investigation. Zawadowicz et al. (2016) demonstrated a method of distinguishing between biogenic and in-organic signals using a different SPMS instrument (PALMS), but it is not yet proven to be transferable to other instruments designs.

## 5 Conclusions

Using a combination of novel sub-composition and crystal structure analysis, we present a detailed characterization of the mineralogy and mixing state of the fine fraction ($< 2.5\mu m$) of North African mineral dust. Differences in clay mineralogy on a continental scale are detectable, with Moroccan dust high in ISCM and Sahelian dust dominated by kaolinite like minerals. In most cases, the distribution of mineralogical composition is more continuous than clustered in distinct phases, and only laboratory suspended river sediment from the Anti-Atlas contained a distinct mode of felsic particles in the fine fraction.

The detailed analysis of laboratory suspended dust provided useful reference data for the interpretation of the origins and transport history of ambient dust in the remote North Atlantic. Relatively large numbers of illite rich particles suggests a dust source on the NW margins of the Sahara during the summer, but the mixing state is key to understanding the transport history.





Internally mixed nitrate suggests dust from the NW margins of the Sahara was deposited into the marine boundary layer after transport in the Saharan air layer. In contrast, this nitrate mixing was much reduced during a short dust event, for which back-trajectories suggest direct emission into the marine boundary layer from the West African coast. The relative concentration of organic/biogenic markers did not vary significantly, but had significant local variation in soil samples, making them less useful

for interpreting source and transport history. The origin of these chemical markers and how they change the physiochemical properties of dust particles requires further investigation.

These results show that the SPMS and SEM EDX techniques are complimentary, with SPMS providing a relative indication of mineralogy and mixing state of the principal particle matrix, and SEM EDX providing semi-quantitative elemental composition of all particles, including material that the SPMS may not ablate. Although non-quantitative, the SPMS techniques

are particularly sensitive to the potassium and sodium content of dust particles which makes them suitable for differentiating continental sediments in real time. These measurements show that changes in mineralogy of ambient aerosol particles can be detected with high temporal resolution using SPMS.

The combination of online mineralogy and mixing state of ambient dust has potential for resolving the complexity of dust dust emission, transport and deposition. These examples of aged transported dust should also be useful in obtaining atmo-

spherically relevant test material for laboratory experiments of ice nucleation and radiative properties, but more high resolution measurements at varying distance from potential dust sources are required to fully understand the complexity of atmospheric dust transport.

*Code availability.* TEXT

*Data availability.* TEXT

*Code and data availability.* TEXT

*Author contributions.* TEXT

*Competing interests.* TEXT

*Disclaimer.* TEXT



*Acknowledgements.* ICE-D was supported by the Natural Environment Research Council (grant number NE/M001954/1), during which N.Marsden was supported by a PhD studentship (NERC M113463J). This project/work has received funding from the European Union's Horizon 2020 research and innovation programme through the EUROCHAMP-2020 Infrastructure Activity under grant agreement No 730997. We thank the AIDA engineering and technical team for maintaining and operating the cloud chamber facility. R. Fösig and O.

5    Möhler acknowledge funding by the Deutsche Forschungsgemeinschaft (DFG) through the research unit INUIT (FOR 1525, project MO 668/4-2)K. Kandler and S. E. Hammer acknowledge financial support from the German Research Foundation (DFG grants KA 2280/2-1 and 3-1 as well as FOR 1525).




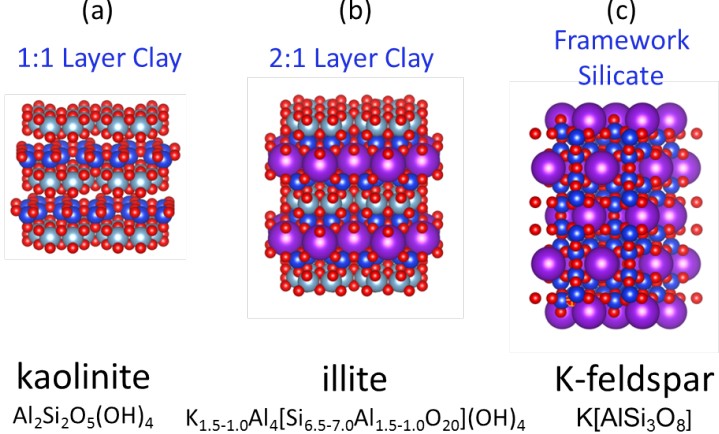

**Figure 1.** The crystal structures of common aluminosilicate minerals (a) 1:1 clay mineral structure of kaolinite, (b) the 2:1 clay mineral structure of illite and (c) the framework structure of k-feldspar.




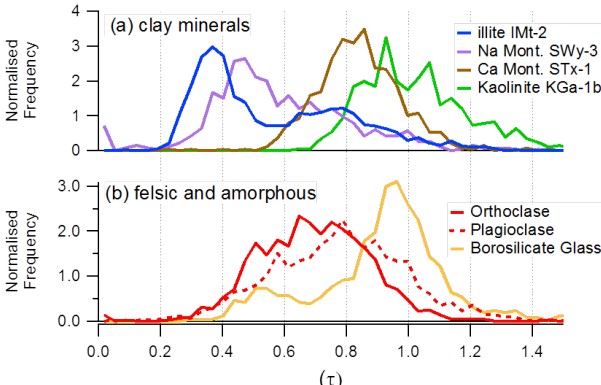

**Figure 2.** Histograms of the peak shift ratio ($\tau$) for particles dispersed in a dust tower. (a) nominally pure clay minerals and (b) feldspars (orthoclase and plagioclase) and borosilicate glass.



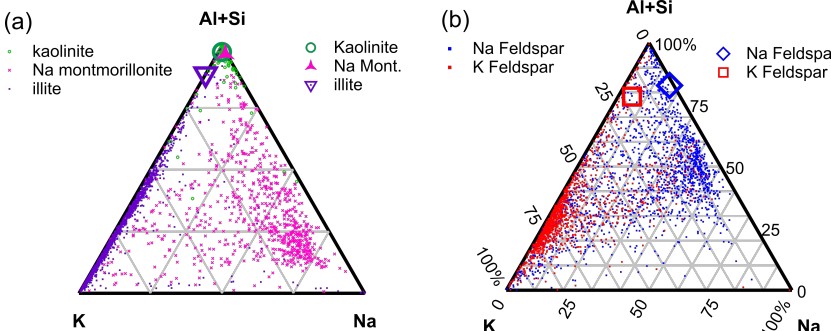

**Figure 3.** Ternary plots of the Al+Si, K and Na content in single particles from lab generated reference material. Each dot represents the composition of a single particle whereas larger icons represent bulk composition by XRF analysis. (a) Nominally pure kaolinite, Na-montmorillonite and illite from the clay mineral society (CMS sample ID KGa-1b, SWy-3 and IMt-3 respectively) (b) feldspar crushed from large crystals.



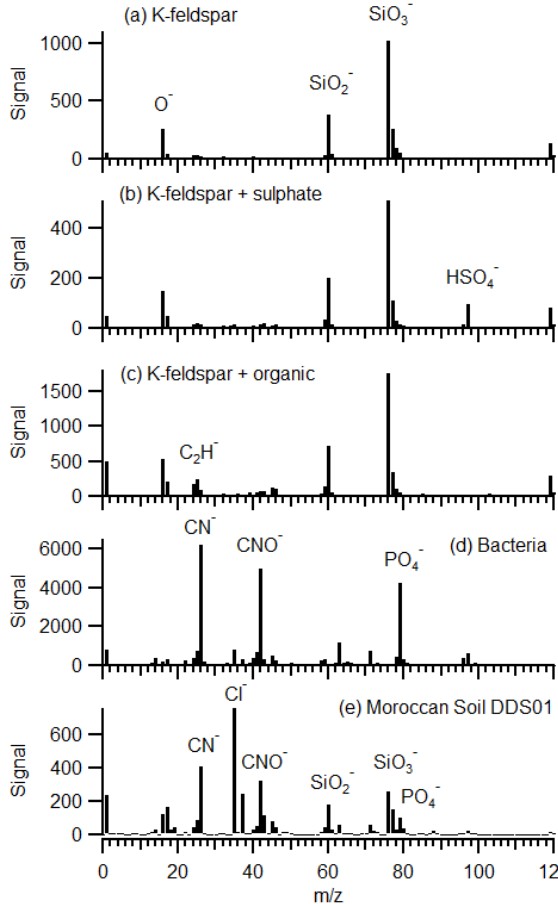

**Figure 4.** Average negative ion mass spectra (200 particles) of re-suspended material in the AIDA chamber during the FIN-1 campaign. (a) pure K-feldspar, (b) K-feldspar with sulfate coating, (c) K-feldspar with organic coating, (d) bacteria and (d) Moroccan soil dust DDS01 during INUIT09.





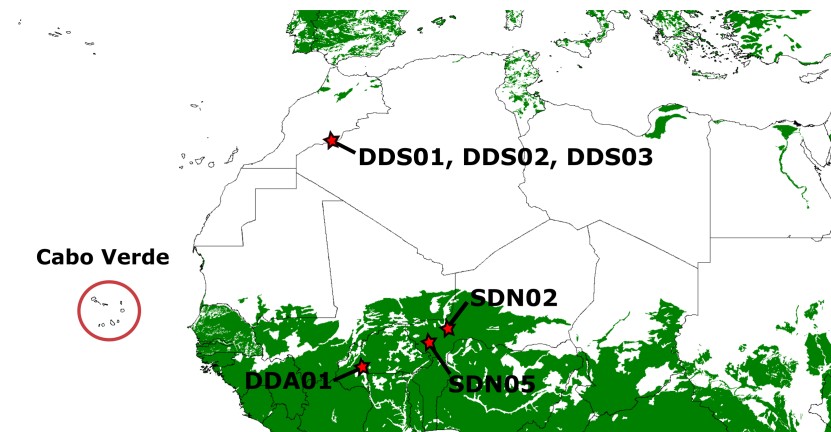

**Figure 5.** Sampling locations of natural soil (stars) and transported dust at Cabo Verde (circle). The green color indicates soils that have a cation ion exchange capacity (CEC) of less than 35 cmol/kg indicating kaolinite rich soil. Produced from the Digital Soil Map of the World.





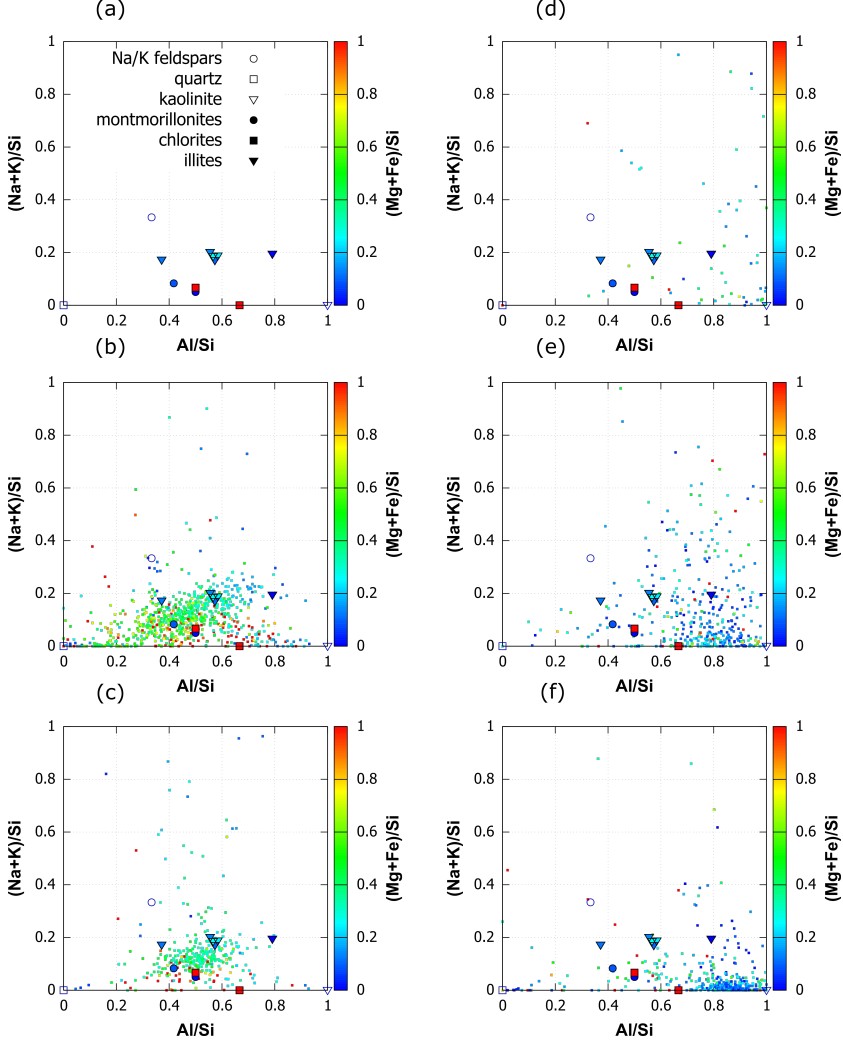

**Figure 6.** Elemental composition of suspended dust samples by SEM EDX analysis. (a) average composition of nominally pure mineral samples, (b) DDS01, (c) DDS02, (d) DDA01, (e) SDN02, (f) SDN05. Ratios are calculated from atomic %




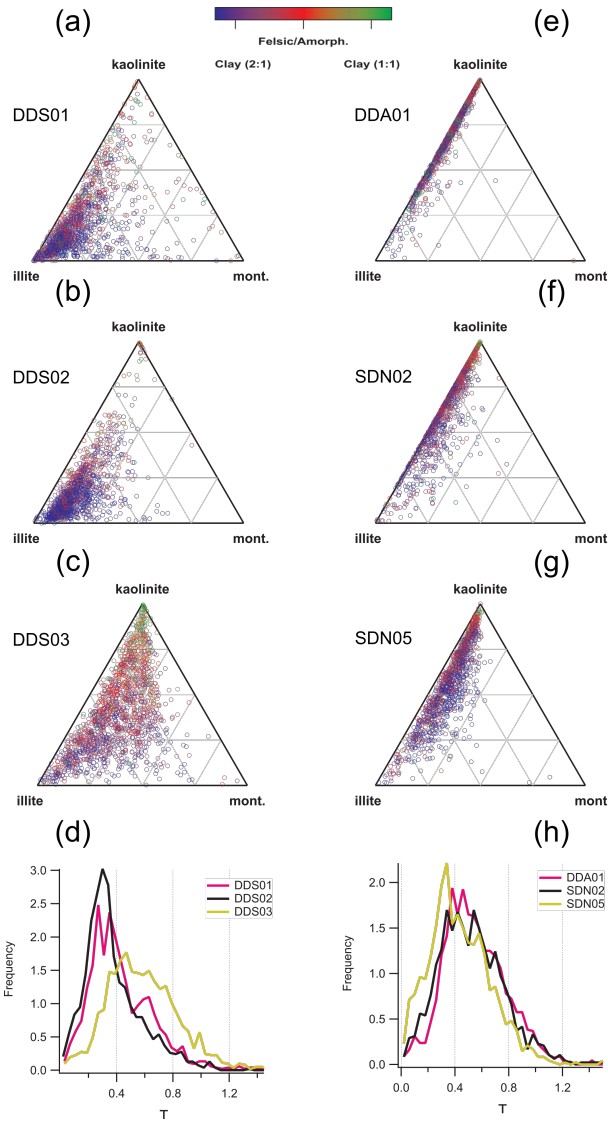

**Figure 7.** Ternary plots of the Al+Si, K and Na content of lab suspended natural soil samples from Morocco (a, b, c) and the Sahel (e, f, g). The color function is proportional to the $\tau$ parameter of crystal structure which is also displayed as a histogram (d, h). Each plot contains 2000 particles)





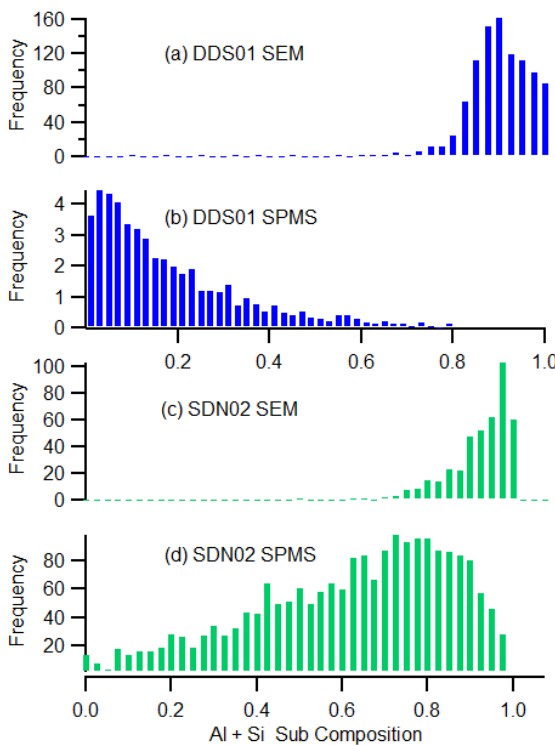

**Figure 8.** Histograms of the Al+Si sub-composition with respect to alkali metals ((Al+Si)/(Al+Si+K+Na)) in single particles of Moroccan (a, b) and Sahelian (c, d) samples using the SEM and SPMS technique.





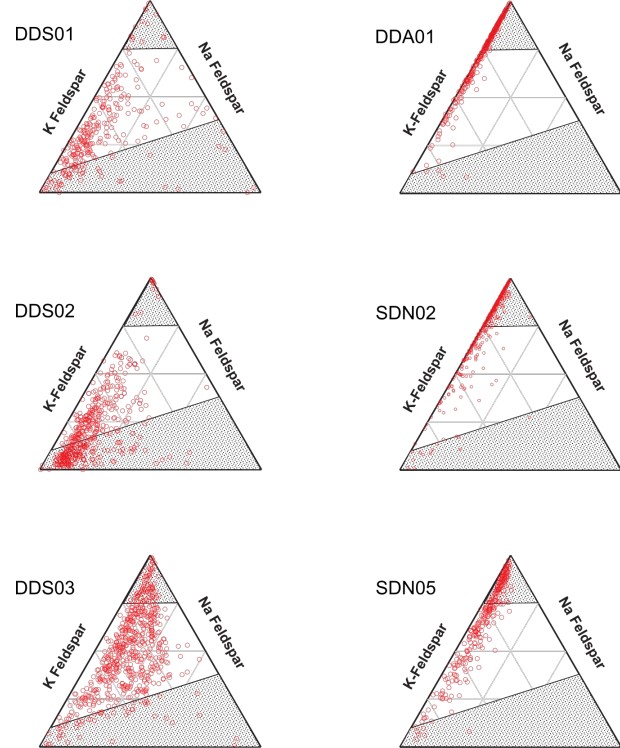

**Figure 9.** Ternary plots of the Al+Si, K and Na content of the felsic/amorphous fraction as determined by crystal structure analysis ($\tau = 0.58 - 0.8$) in lab suspended natural soil samples from Morocco (a, b,c) and the Sahel (d, e, f). The greyed out area represent unlikely composition of feldspar as demonstrated by the fingerprint in Fig. 3.





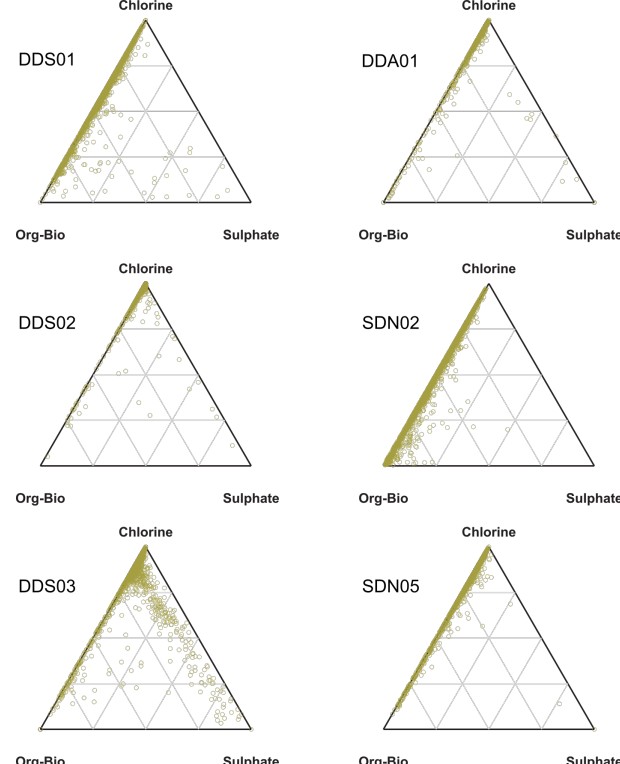

**Figure 10.** Ternary plots of the Cl, organic-biological and sulphate marker peak areas in Moroccan (a, b, c) and Sahelian (c,d,e) re-suspended soil dust.





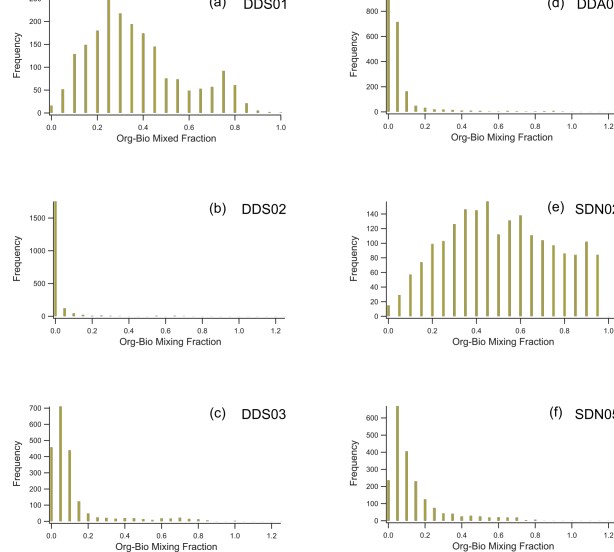

**Figure 11.** Histograms of the normalised Org-Bio sub-composition from single particle of Moroccan (a, b, c) and Sahelian (d, e, f) samples.

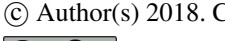



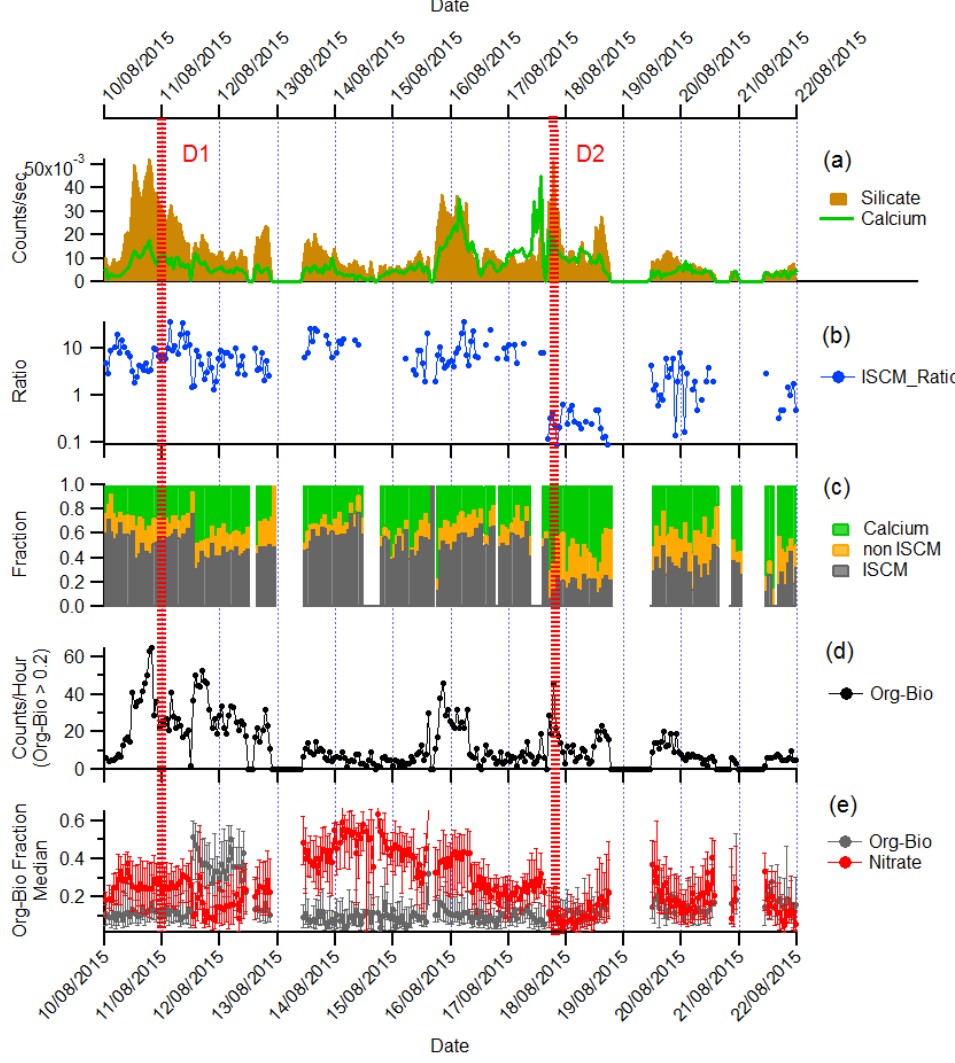

**Figure 12.** Time series (1 hour resolution) of ambient dust properties measured at Cabo Verde during the ICE-D campaign by LAAPTOF SPMS. (a) concentration of silicate and calcium rich particles. (b) Hourly average ratio of ISCM to non-ISCM minerals determined by crystal structure, (c) Hourly particle number fraction of calcium, ISCM and non-ISCM particles. (d) Hourly concentration of silicate particles with organic-biological sub-composition > 0.2. (e) Hourly median Org-Bio and nitrate sub-composition marker quantity. Error bar is 25th and 75th percentile.





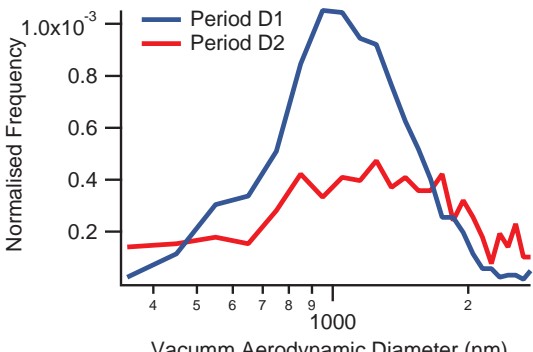

**Figure 13.** Vacuum aerodynamic size distributions of silicate particles in dust event D1 and D2.





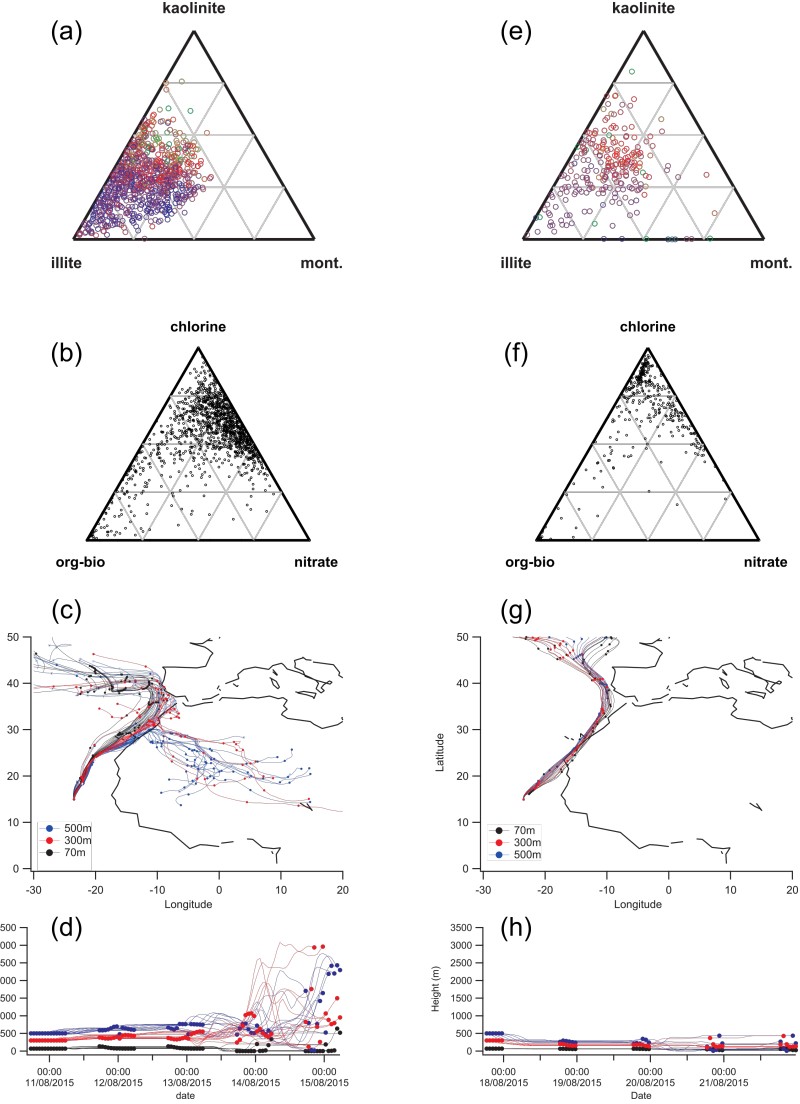

**Figure 14.** Sub-composition and transport history of dust events D1 (a-d) and D2 (e-f), showing mineralogy (a and e), mixing state (d and f) and HYSPLIT back trajectory analysis (c,d,g,h).



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
