# Peer review of "Mineralogy and mixing state of North African mineral dust by on-line single-particle mass spectrometry."

_Atmospheric Chemistry and Physics, 2018_

## Referee Comment (RC1) · Anonymous Referee #3 · 15 Oct 2018

First, I sincerely apologize for the delay of my revision.

The paper by Marsden et al. analyses the mineralogy and mixing state of North African mineral dust by applying a novel on line single particle mass spectrometry technique (LAAPTOF instrument) and comparing it to results by the more classical scanning electron microscopy analysis (SEM). Data were collected for natural dust observed at Cape Verde during the ICE-D campaign and from experiments on re-suspended dust in the AIDA chamber facility.

Results allow to show the performances of the LAAPTOF in estimating the mineralogical signature and mixing state of particles with different origins as measured at Cape

[Figure]

Verde and from chamber data. The paper provides a thorough analysis of the chemical composition and mineralogy. Discussion of possible atmospheric implications of taking advantage of measurements presented in this paper is provided.

This is a very valuable work presenting new results from a novel technique to study the mineralogy of dust aerosols and its applications. While I do not have major comments on the content of the paper itself, which I find of very high quality, I have some doubts on the presentation of data and methods and discussion of results. In fact I find the paper too much detailed in some specific aspects, in particular the full section 2.1 seems confusing and it is not clear what is method, results or discussion. Probably the presentation of such part should be modified. At the same time some specific details that would be required are not presented in Section 2 (more details in time resolution, uncertainties of the used techniques). Despite the reference is done to another methodological paper, these information should be done also in the present work for consistency. The "Results" section also deserves to be discussed probably in a "lighter" way to be more accessible also for a non-advanced reader, and probably merged to the discussion in Sect. 2.1.1-2.1.3. In conclusion, I think the work deserves publication on ACP but only after the presentation is better organized, probably shortened in some parts. I propose some comments to the authors in the following together with some minor comments.

Detailed comments

Abstract, line 6-7: specify the temporal resolution required

Abstract, line 19-20: please clarify the sentence

Page 2, line 8: "observations"

Page 2, lines 16-22: this section is confusing; you speak about soil to aerosol partitioning and atmospheric ageing, but this is not very clear

Page 2, lines 23-28: also this section is confusing since it mixes considerations on the

temporal and spatial resolution of dust mineralogy

Page 3, line 22: the reference by Kok et al. (2017) is not appropriate here since the Kok et al. paper is mostly on the effect of particle size but not mineralogy

Section 2.1 please add information on the temporal resolution and measurement uncertainties

Sect. 2.1.1, 2.2.2, 2.1.3: please reorganize these sections to clarify the methodological aspects, while the aspects mostly linked to discussion of data should be moved to Sect. 3 ("Results")

Sect. 2.2: which is the impact of the differences in the sampled size distribution compared to LAAPTOF?

Sect. 2.4: the vicinity of the measurement site to the airport runway has an effect on measurements?

Page 10, line 8: "to analyse"

Results and Discussion, sections 3 and 4: I invite the authors to consider if some of the aspects in Sections 2.1 should not be moved here and also if the whole presentation of results and the discussion should not be reduced a little bit in length and also made more readable for a non-advanced reader. The text is fact very dense and complicated in some points in my opinion and it could be probably simplified.

---

## Referee Comment (RC2) · Anonymous Referee #1 · 18 Oct 2018

Marsden et al. use laser desorption/ionization single-particle mass spectrometry to analyze source mineral dust particles, and ambient dust-laden aerosol sampled in North Africa. They apply their recently developed mass spectral analysis method to distinguish between different silicate minerals, based on the time delay that the relevant ions are detected in the MS. This time delay relates to the crystal structure of the different mineral phases. Distinguishing between different specific mineral phases or families in individual particles is a serious analytical challenge. The results presented here are quite novel, despite the various issues related to the semi-quantitative nature of the analysis, only being able to identify or distinguish some of the major mineral phases, and not achieving complete separation between different phases (the spread in the

delay time for different "pure" minerals is quite large with significant overlap). The analysis presented here could be more fairly put in context of SP-MS analysis focused on mineral dust that has come before, while still highlighting the advance made here regarding between distinguishing between mineral phases.

I think the importance of single-particle measurements of mineral phases should be developed more in the introduction to better motivate the work. This is a major challenge that limits our understanding not just of sources and transport of mineral dust, but also of their critical properties. Knowledge of mineralogy (as opposed to just elemental composition) is necessary to understand chemical reactivity, and is crucial to predict the mineral particles' ice nucleation properties.

The manuscript was often hard to follow, sometimes written more like a lab report with concepts and terms suddenly introduced with no explanation or definition. It was especially difficult to keep track of where the various geographical locations that are mentioned repeatedly actually are. Many of the figures are not designed that well, do not have captions that adequately explain the figure, and could be improved in their clarity. Several sections often start with just one sentence before the sub0section starts, an odd way to start a section. Many typos and syntax errors abound, and many references cite ACPD versions of papers instead of the ACP version; the manuscript needs to be carefully proofread and improved. The analysis and results presented here will certainly be of interest to the ACP community. This manuscript should be acceptable for publication in ACP once the authors fully address the questions raised and improve the manuscript's clarity.

Introduction: (Jickells et al., 2005) is an excellent review of the importance of atmospheric mineral dust for the oceans and biogeochem.

Pg 3/line 14: It is not really accurate to refer to the "IN fraction". A fraction of particles can be IN active at a /specific/ temperature. Most mineral phases are IN active at some mixed-phase cloud temperature, even weak ice nucleants such as quartz. Perhaps rephrase to refer to the ice nucleation properties of mineral dust particles (which requires knowledge of the specific mineral phases present).

Pg 3/line 16: When discussing ice nucleation it is important to refer to the ice nucleation mode being referred to. While strong acids and SOA added to dust particles can impair deposition freezing, they do not seem to interfere with immersion freezing. Presumably the condensate or reaction product dissolves off in the droplet (Niedermeier et al., 2011; Reitz et al., 2011; Sullivan et al., 2010a, 2010b)

3/22-29: This paragraph rather discounts the many important observations that have been made regarding individual mineral dust composition, aging, and reactivity achieved through offline electron/x-ray microscopy, for example: (Hwang and Ro, 2006; Jeong and Chun, 2006; Krueger et al., 2003, 2004; Ro et al., 2005; Sobanska et al., 2012; Tobo et al., 2009; Zhang et al., 2003).

4/4: "SPMS...but not differentiate mineralogy." This is an entirely inaccurate statement. Indeed later in the paper prior use of SP-MS to differentiate mineralogy is presented. But even there much of the closely relevant prior studies using SPMS to analyze dust mineralogy is not discussed or cited. The first detailed look at how dust mineralogy influences the chemistry of atmospheric dust, as well as particle size and mixing state analysis, was presented by (Sullivan et al., 2007a) in ACP. That analysis found that Ca-rich particles accumulated more nitrate and chloride, while Fe/Al-rich particles took up more sulfate, for example. The Sullivan et al. paper cited here focuses on chlorine chemistry in mineral dust (Sullivan et al., 2007b), which is certainly relevant, but the analysis presented in the ACP paper came first and is more closely relevant to the analysis presented here. There is another paper that focuses on organic acids in mineral dust particles using SP-MS (Sullivan and Prather, 2007). I mention this series of papers that use the ATOFMS as I believe they were the first to really analyze in detail the mixing state and mineralogy of individual dust particles using SP-MS.

Sect. 2.1: More details regarding the particle detection system should be provided. Is

Interactive
comment

this a custom non-standard setup for the LAAPTOF? Nowhere is the typical particle detected fraction (as a function of particle size) presented. This is critical information as the sub-population of total ambient particles actually detected by the instrument can significantly bias the measurements and analysis. The particle detection rate also governs the ability to observe changes in particle composition and sources over short timescales, something that is focused on here.

The relevant parameter for LDI is the laser energy power density (W/cm2). What is the laser diameter when it intersects the particle, so this quantity can be reported?

5/5: Just referring the reader to another paper for relevant information is not satisfactory. Please provide a clear summary here regarding the relevant performance characteristics of this LAAPTOF configuration that influences what fraction of particles are actually detected, put in the context of the aerosol populations analyzed here and their associated properties that govern particle sample and detection.

5/6: The LAAPTOF is quite similar in design to prior SP-MS instruments. I do not see why the authors think they can only refer to the few prior LAAPTOF papers that have demonstrated quantitative particle analysis. There is a large body of SP-MS work demonstrating the semi-quantitative capabilities, and even truly quantitative analysis if it is done carefully with calibration (Bhave et al., 2002; Fergenson et al., 2001; Gross et al., 2000). The use of SP-MS to determine heterogeneous kinetics is perhaps the best example of quantitative analysis (Saul et al., 2006; Sullivan et al., 2009).

6/1: It would be useful to expand on this weathering of minerals and how that relates to this analysis. Later in the paper feldspars are discussed as indicating less aged/weathered mineral particles, and I think that idea comes from the information presented here, but the connection is not clear. Feldspars can also be converted to the amorphous clays by acid attack, right? (Wex et al., 2014)

6/15: It is not clear if this analysis works for all silicates such as quartz, which is a very common mineral. Please discuss.

6/23: The poor separation of K+ and Ca+ is an important limitation in this analysis. Please explain the causes of this. Is it specific to the LAAPTOF's configuration? While space charge effects that degrade ion resolution are common in LDI-MS of ion-rich mineral particles, usually K and C can be resolved. The low ionization energy of K can also produce an overly broad ion peak that extends past m/z 39.5 and into Ca at m/z 40. Is this the issue here? Is the LAAPTOF's MS too poor in resolution to resolve K and Ca?

6/31: The analysis discussed here is of a semi-quantitative (relative) nature, so I don't see why the authors refer to it as "non quantitative". That has a quite different meaning.

Sect. 2.1.2: At the end there really needs to be a summary of what mineral phases can be included in this type of analysis, and which can be distinguished. I was pretty confused as to what mineral phases (such as quartz, or carbonates) are and are not included in the analysis performed here. A good discussion of how reliably the various mineral phases (that can be analyzed) can be discriminated from each other is also needed. Please be as quantitative there as possible.

7/7: Reactions with ozone will just convert O3 to O2 on dust, and not add any material.

7/10: Usually see other more specific ion markers for organics in dust using SP-MS. m/z +43 is common for oxidized organics, and negative ions often have fragments from organic acids (Silva and Prather, 2000; Sullivan and Prather, 2007). Please discuss why only very small organic fragments are observed here. The LDI laser pulse energy of 3-5 mJ is rather high and perhaps caused extensive fragmentation. Also, while organics can "char" to EC ions, C2+ could also come from black carbon that was mixed with the particle. I would be wary of using C2+ to identify organic carbon, it is a generic carbonaceous (OC + BC) marker.

7/19: More unique biological ion markers are often observed, such as phosphates, as well as the rather ubiquitous (Murphy et al., 2006) CN- and CNO- ions, if you look at the spectra presented from ATOFMS analysis for example (Creamean et al., 2013; Pratt et

al., 2009; Sultana et al., 2017).

7/23: The use of chloride in the analysis is confusing and needs more discussion. Chloride is not only a part of primary mineral components but can also accumulate in dust via transport and uptake of HCl(g) etc. Please better explain the purpose of using Cl- in the analysis, and how that would be affected by secondary sources of Cl in the dust particles.

8/14: Rationing the signals to Si seems to follow the work of Sullivan et al. that normalized to Al. The use of Si is likely more appropriate, especially in this work's focus on silicate minerals. There is also other work that introduced ternary analysis to understand mineral dust composition and aging (Krueger et al., 2003, 2004; Laskin et al., 2005; Yuan et al., 2004)

Sect. 2.3: Odd to just have one sentence here. It also inaccurately implies that distinguishing between different minerals using SP-MS has never been done before.

9/14: what are these, and how to they relate to mineral composition? "fluvisols (50%), associated with yermosols (20%), regosols (20%) and solonchaks (10%)."

9/15: It took me awhile to realize that these are the locations that each sample was taken from. Would probably be better in a Table.

Sect. 3: Again odd to just have one sentence, and this one has many syntax errors/typos.

11/3: greater sensitivity to alkali metals is due to low ionization energy? Please explain. There is too much expected expert knowledge that non SP-MS users will not necessarily know.

11/7: where were these soil samples from? Also North Africa?

11/12: felsic means felspars?

11/19: This is a good example of where a brief reminder of what this "Anti-Atlas" location is would be useful.

12/13: This is why a proper discussion of the LAAPTOF's detection efficiency versus particle size (for dust particles) is needed. This suggests that the instrument is only detecting 1% of total silicate particles. How much of this is just due to particle size transmission issues, versus the instrument's actually particle hit percentage?

12/23: Uptake of HCl by dust observed using SP-MS was a focus of (Sullivan et al., 2007b), which you cite yet oddly do not discuss when very relevant here.

13/2: Tenerife?

13/31: What metric is "> 0.2"? Ion peak area? How do you decide if a particle has a "significant fraction (> 0.2)"?

14/11: This dust mobilization refers to the emissions of dust or its transport? Confusing.

14/23: Can also have nitrate from coagulation with ammonium nitrate, uptake of N2O5, etc. (Korhonen et al., 2003; Sullivan et al., 2007a; Zhang et al., 2000). Can't conclude it is all from HNO3.

14/25: By biogenic source are you referring to sulfate derived from DMS? Explain.

15/11: Some key references to whole rock geochemical analysis would be nice.

15/13: Important paper on ternary analysis: (Yuan et al., 2004)

15/14: Matrix effects in LDI are mentioned repeatedly but never explained or discussed. Non-experts will be unfamiliar with this important effect.

15/28: Is the reader supposed to know where Praia Cabo Verde is? If so I have forgotten so a reminder would be useful.

Page 15: Fractionation of mineralogy versus particle size during transport is never discussed (Arimoto et al., 2004; Gong et al., 2003; Mori et al., 2003). Nor is the size distribution of the different mineral dust types presented, which is a real oversight.

15/34: Quantitative information regarding by how much the mineral composition can change over just one hour would be very useful here. How significant a change can occur?

16/20: What are these accompanying aircraft sorties? Another example of new ideas that are just sort of thrown out there without proper introduction or explanation.

16/22: As mentioned above, there are more and more specific biological ion markers to use than just CN- and CNO-. Also please summarize the analysis method developed by Zawadowicz, so it can be better understood why it might not be transferrable from the PALMS to LAAPTOF. If both use the same LDI laser wavelength it is likely transferrable.

Conclusions: Mentioning the timescale and magnitude under which changes in mineralogy occur would be good information to include here. That seems to be one of the major findings from the ambient measurements.

17/9: Really semi-quantitative, not non quantitative. Almost all the analysis presented here is quantitative-based, not just qualitative.

17/14: Understanding individual dust mineralogy is also important for understanding reactivity!

Fig. 1: Explaining the color code for the ions in the caption would be useful.

Fig.2: Should cite your prior paper in the caption where this method was developed so the connection is clear.

Fig. 4: The phosphate marker I mentioned above for biological is evident here, why is it not also used? Granted there are mineral sources of phosphate, but it could be used in combination with CN- and CNO-.

Fig. 6: I could not find the point for pure quartz. Explain in caption that big symbols are for reference samples, and make the symbol line thicker so they are easier to see. Also add the sample name to the top of each plot as in Fig. 7.

[Figure]

Fig. 7: Please explain what this means: "The color function is proportional to the $\tau$ parameter of crystal structure which is also displayed as a histogram"

Fig. 10: Hard to see symbols. Make lines thicker and use a different color.

References: Many cite the ACPD version instead of the ACP one. Please correct.

Pg 2/line 21: "affects" not effects.

Cited References:

Arimoto, R., Zhang, X. Y., Huebert, B. J., Kang, C. H., Savoie, D. L., Prospero, J. M., Sage, S. K., Schloesslin, C. A., Khaing, H. M. and Oh, S. N.: Chemical composition of atmospheric aerosols from Zhenbeitai, China, and Gosan, South Korea, during ACE-Asia, J. Geophys. Res., 109(D19), doi:doi:10.1029/2003JD004323, 2004. Bhave, P. V, Allen, J. O., Morrical, B. D., Fergenson, D. P., Cass, G. R. and Prather, K. A.: A field-based approach for determining ATOFMS instrument sensitivities to ammonium and nitrate, Environ. Sci. Technol., 36(22), 4868–4879, 2002. Creamean, J. M., Suski, K. J., Rosenfeld, D., Cazorla, A., Demott, P. J., Sullivan, R. C., White, A. B., Ralph, F. M., Minnis, P., Comstock, J. M., Tomlinson, J. M. and Prather, K. A.: Dust and biological aerosols from the Sahara and Asia influence precipitation in the western U.S., Science, 339(6127), 1572–1578, doi:10.1126/science.1227279, 2013. Fergenson, D. P., Song, X. H., Ramadan, Z., Allen, J. O., Hughes, L. S., Cass, G. R., Hopke, P. K. and Prather, K. A.: Quantification of ATOFMS data by multivariate methods, Anal. Chem., 73(15), 3535–3541, 2001. Gong, S. L., Zhang, X. Y., Zhao, T. L., McKendry, I. G., Jaffe, D. A. and Lu, N. M.: Characterization of soil dust aerosol in China and its transport and distribution during 2001 ACE-Asia: 2. Model simulation and validation, J. Geophys. Res., 108(D9), doi:doi:10.1029/2002JD002633, 2003. Gross, D. S., Galli, M. E., Silva, P. J. and Prather, K. A.: Relative sensitivity factors for alkali metal and ammonium cations in single particle aerosol time-of-flight mass spectra, Anal. Chem., 72(2), 416–422, 2000. Hwang, H. and Ro, C.: Direct observation of nitrate and sulfate formations from mineral dust and sea-salts using low-Z particle electron probe X-ray microanalysis, Atmos. Environ., 40(21), 3869–3880, doi:10.1016/j.atmosenv.2006.02.022, 2006. Jeong, G. Y. and Chun, Y.: Nanofiber calcite in Asian dust and its atmospheric roles, Geophys. Res. Lett., 33(24), L24802, doi:10.1029/2006GL028280, 2006. Jickells, T. D., An, Z. S., Andersen, K. K., Baker, A. R., Bergametti, G., Brooks, N., Cao, J. J., Boyd, P. W., Duce, R. A., Hunter, K. A., Kawahata, H., Kubilay, N., laRoche, J., Liss, P. S., Mahowald, N., Prospero, J. M., Ridgwell, A. J., Tegen, I. and Torres, R.: Global iron connections between desert dust, ocean biogeochemistry, and climate, Science (80-. )., 308(5718), 67–71, 2005. Korhonen, H., Napari, I., Timmreck, C., Vehkamaki, H., Pirjola, L., Lehtinen, K. E. J., Lauri, A. and Kulmala, M.: Heterogeneous nucleation as a potential sulphate-coating mechanism of atmospheric mineral dust particles and implications of coated dust on new particle formation, J. Geophys. Res., 108(D17), doi:doi:10.1029/2003JD003553, 2003. Krueger, B. J., Grassian, V. H., Iedema, M. J., Cowin, J. P. and Laskin, A.: Probing heterogeneous chemistry of individual atmospheric particles using scanning electron microscopy and energy-dispersive X-ray analysis, Anal. Chem., 75(19), 5170–5179, 2003. Krueger, B. J., Grassian, V. H., Cowin, J. P. and Laskin, A.: Heterogeneous chemistry of individual mineral dust particles from different dust source regions: the importance of particle mineralogy, Atmos. Environ., 38(36), 6253–6261, 2004. Laskin, A., Wietsma, T. W., Krueger, B. J. and Grassian, V. H.: Heterogeneous chemistry of individual mineral dust particles with nitric acid: A combined CCSEM/EDX, ESEM, and ICP-MS study, J. Geophys. Res., 110(D10), D10208, doi:doi:10.1029/2004JD005206, 2005. Mori, I., Nishikawa, M., Tanimura, T. and Quan, H.: Change in size distribution and chemical composition of kosa (Asian dust) aerosol during long-range transport, Atmos. Environ., 37(30), 4253–4263, 2003. Murphy, D. M., Cziczo, D. J., Froyd, K. D., Hudson, P. K., Matthew, B. M., Middlebrook, A. M., Peltier, R. E., Sullivan, A., Thomson, D. S. and Weber, R. J.: Single-particle mass spectrometry of tropospheric aerosol particles, J. Geophys. Res., 111(D23), doi:doi:10.1029/2006JD007340, 2006. Niedermeier, D., Hartmann, S., Wex, H., Clauss, T., Kiselev, A., Sullivan, R. C., DeMott, P. J., Petters, M. D., Reitz, P., Schneider, J., Mikhailov, E., Reimann, B., Bundke, U., Stetzer, O., Sierau, B.,

Shaw, R. A., Mentel, T. F. and Stratmann, F.: Experimental study of the role of physic-ochemical surface processing on the IN ability of mineral dust particles, Atmos. Chem. Phys., 11, 11131–11144, 2011. Pratt, K. A., DeMott, P. J., French, J. R., Wang, Z., Westphal, D. L., Heymsfield, A. J., Twohy, C. H., Prenni, A. J. and Prather, K. A.: In situ detection of biological particles in cloud ice-crystals, Nat. Geosci., 2(6), 397–400, doi:10.1038/ngeo521, 2009. Reitz, P., Spindler, C., Mentel, T. F., Poulain, L., Wex, H., Mildenberger, K., Niedermeier, D., Hartmann, S., Clauss, T., Stratmann, F., Sullivan, R. C., DeMott, P. J., Petters, M. D., Sierau, B. and Schneider, J.: Surface modification of mineral dust particles by sulphuric acid processing: implications for ice nucleation abilities, Atmos. Chem. Phys., 11, 7839–7858, doi:10.5194/acp-11-7839-2011, 2011. Ro, C. U., Hwang, H., Chun, Y. and Van Grieken, R.: Single-particle characterization of four "Asian Dust" samples collected in Korea, using low-Z particle electron probe X-ray microanalysis, Environ. Sci. Technol., 39(6), 1409–1419, 2005. Saul, T. D., Tolocka, M. P. and Johnston, M. V: Reactive uptake of nitric acid onto sodium chloride aerosols across a wide range of relative humidities., J. Phys. Chem. A, 110, 7614–7620, 2006. Silva, P. J. and Prather, K. A.: Interpretation of mass spectra from organic compounds in aerosol time-of-flight mass spectrometry, Anal. Chem., 72(15), 3553–3562, 2000. Sobanska, S., Hwang, H., Choël, M., Jung, H.-J., Eom, H.-J., Kim, H., Barbillat, J. and Ro, C.-U.: Investigation of the Chemical Mixing State of Individual Asian Dust Particles by the Combined Use of Electron Probe X-ray Microanalysis and Raman Microspectrometry., Anal. Chem., 84(7), 3145–54, doi:10.1021/ac2029584, 2012. Sullivan, R. C. and Prather, K. A.: Investigations of the Diurnal Cycle and Mixing State of Oxalic Acid in Individual Particles in Asian Aerosol Outflow, Environ. Sci. Technol., 41(23), 8062–8069, doi:10.1021/es071134g, 2007. Sullivan, R. C., Guazzotti, S. A., Sodeman, D. A. and Prather, K. A.: Direct observations of the atmospheric processing of Asian mineral dust, Atmos. Chem. Phys., 7(5), 1213–1236, doi:10.5194/acp-7-1213-2007, 2007a. Sullivan, R. C., Guazzotti, S. A., Sodeman, D. A., Tang, Y., Carmichael, G. R. and Prather, K. A.: Mineral dust is a sink for chlorine in the marine boundary layer, Atmos. Environ., 41(34), 7166–7179, doi:10.1016/j.atmosenv.2007.05.047,

2007b. Sullivan, R. C., Moore, M. J. K., Petters, M. D., Kreidenweis, S. M., Roberts, G. C. and Prather, K. A.: Timescale for hygroscopic conversion of calcite mineral particles through heterogeneous reaction with nitric acid, Phys. Chem. Chem. Phys., 11(36), 7826, doi:10.1039/b904217b, 2009. Sullivan, R. C., Miñambres, L., DeMott, P. J., Prenni, A. J., Carrico, C. M., Levin, E. J. T. and Kreidenweis, S. M.: Chemical processing does not always impair heterogeneous ice nucleation of mineral dust particles, Geophys. Res. Lett., 37(24), doi:10.1029/2010GL045540, 2010a. Sullivan, R. C., Petters, M. D., DeMott, P. J., Kreidenweis, S. M., Wex, H., Niedermeier, D., Hartmann, S., Clauss, T., Stratmann, F., Reitz, P., Schneider, J. and Sierau, B.: Irreversible loss of ice nucleation active sites in mineral dust particles caused by sulphuric acid condensation, Atmos. Chem. Phys., 10, 11471–11487, 2010b. Sultana, C. M., Al-Mashat, H. and Prather, K. A.: Expanding Single Particle Mass Spectrometer Analyses for the Identification of Microbe Signatures in Sea Spray Aerosol, Anal. Chem., 89(19), 10162–10170, doi:10.1021/acs.analchem.7b00933, 2017. Tobo, Y., Zhang, D. Z., Nakata, N., Yamada, M., Ogata, H., Hara, K. and Iwasaka, Y.: Hygroscopic mineral dust particles as influenced by chlorine chemistry in the marine atmosphere, Geophys. Res. Lett., 36, L05817, doi:10.1029/2008gl036883, 2009. Wex, H., DeMott, P. J., Tobo, Y., Hartmann, S., Rösch, M., Clauss, T., Tomsche, L., Niedermeier, D. and Stratmann, F.: Kaolinite particles as ice nuclei: learning from the use of different kaolinite samples and different coatings, Atmos. Chem. Phys., 14(11), 5529–5546, doi:10.5194/acp-14-5529-2014, 2014. Yuan, H., Rahn, K. A. and Zhuang, G.: Graphical techniques for interpreting the composition of individual aerosol particles, Atmos. Environ., 38(39), 6845–6854, 2004. Zhang, D. Z., Shi, G. Y., Iwasaka, Y. and Hu, M.: Mixture of sulfate and nitrate in coastal atmospheric aerosols: individual particle studies in Qingdao (36 degrees 04 ' N, 120 degrees 21 ' E), China, Atmos. Environ., 34(17), 2669–2679, 2000. Zhang, D. Z., Iwasaka, Y., Shi, G. Y., Zang, J. Y., Matsuki, A. and Trochkine, D.: Mixture state and size of Asian dust particles collected at southwestern Japan in spring 2000, J. Geophys. Res., 108(D24), doi:doi:10.1029/2003JD003869, 2003.

---

## Referee Comment (RC3) · Anonymous Referee #2 · 23 Oct 2018

This is a very well-written paper analyzing single particle measurements of silicate dust for pure laboratory reference material, soils collected from dust-productive regions of Africa and in-situ measurements at Cabo Verde. This is both a timely topic and the analysis of single particle spectra is rigorous, making use of laboratory reference material for comparison with field measurements.

Overall, I recommend the paper for publication in ACP after addressing the major and minor concerns below.

Major comments: I would like to see more discussion of uncertainties in these measurements. In particular, the manuscript mentions that the LAAPTOF instrument tends

[Figure]

to undercount silicate particles (page 12, line 13). Is there any evidence that it could undercount selectively and thus introduce a bias into the results as presented? In other words, are there any chemical biases in the way LAAPTOF detects silicate-rich particles?

In the analysis of dust mixing state, chlorine, CN- and CNO- (termed "org-bio") and sulfate (later nitrate) were chosen as mixing state markers. Comparing between soils collected from the ground and particles analyzed in-situ for these particular components is complicated because of atmospheric processing, but the text seems to draw an equivalence here. For example, CN- and CNO- might indicate a biological or biogenic fraction for soils, but in situ they are much more likely to arise during atmospheric processing and using them as biological markers leads to large overestimates. Similarly, the large chlorine fractions at Cabo Verde are largely expected because of marine influence at that sampling location, but their origin is likely very different in the laboratory soils collected in-land.

Minor comments: In section 2.1.1, large parts of the text (especially first two paragraphs read like introductory material instead of methods.

Page 8, line 17: Figure 6 is called out in the text before Figures 4 or 5.

General comment for the methods section: please indicate the number of single particles analyzed in laboratory and field studies.

Results, first sentence: "...we choose to analysis..." should be we chose to analyze? Please clarify.

Section 3.1, line 12: the authors say that vast majority of particles contained silicate markers, but then they also say that all particles contained some silicate minerals. This is a bit vague.

Page 11, lines 2-5: I am not sure I follow the reasoning here and in Figure 8. For the Moroccan sample, the two techniques seem to be showing the exact opposite

composition.

Section 3.2, line 9: Why were peaks shifted in positive and not negative spectra?

Page 15, line 10: "Sub-compositional analysis is a techniques" – should be technique.
* * *

---

## Author Comment (AC1) · 21 Jan 2019

**Detailed Response to Anonymous Referee #3**

Abstract, line 6-7: specify the temporal resolution required
The sentence now states:
but also because of the lack of an efficient method to report the mineralogy and mixing state of single particles with a time resolution comparable to atmospheric processes lasting a few hours or less.

Abstract, line 19-20: please clarify the sentence
Now reads:
In most cases, the difference in composition between particles within a sample was continuous, rather than a collection of particles with discrete mineral phases.

Page 2, line 8: "observations"
Corrected.

Page 2, lines 16-22: this section is confusing; you speak about soil to aerosol partitioning and atmospheric ageing, but this is not very clear
Now reads:
the abundance of mineral phase has a strong grain size dependence, with quartz occurring in the coarse fraction and clay minerals dominating the fine fraction, but the size distribution is modified during emission \citep{Perlwitz2015}, so that ratios of mineral phases in the lofted mineral dust aerosol may not be completely representative of that of the source soil.

Page 2, lines 23-28: also this section is confusing since it mixes considerations on the temporal and spatial resolution of dust mineralogy

Removed the reference to temporal resolution in this paragraph.

Page 3, line 22: the reference by Kok et al. (2017) is not appropriate here since the Kok et al. paper is mostly on the effect of particle size but not mineralogy
Removed reference.

Section 2.1 please add information on the temporal resolution and measurement un-Certainties
We have clarified the potential sampling rate in the following sentence:
The instrument is capable of providing size resolved composition measurements for up to 200 particles per second in the size range approx. $0.4-2.5\mu m$

Added the following paragraph regarding measurement efficiencies.
Laboratory evaluation of the fiber-coupled laser system indicate that the detection efficiency peaks at 0.25 with spherical particles \cite{Marsden2016a}, but the overall efficiency of the instrument also depends on ablation efficiency with respect to particle composition. In a study of nominally pure mineral samples,

\cite{Marsden2017} reported the number of optically detected particles that produced a mass spectra (i.e. ablation efficiency or hit rate) of 0.29 and 0.14 for illite and kaolinite respectively, but was also dependent on the amount of impurities such as Titanium. Furthermore, from the authors own experience, it likely that pure quartz may have an ablation efficiency close to zero and is not considered in our analysis, but is unlikely to be a major component in the fine fraction in any case. The potential maximum overall efficiency of the LAAPTOF measurement of clay mineral ranges from 0.0725 for pure spherical particles particles of illite, to 0.035 for pure spherical particles of kaolinite. The exact efficiency of the instrument is not known in most situations because the size, shape and composition of the particles would have to be known a priori for accurate calibration.

We have also included the following in the conclusions section:
Despite the fact that the technique provides incomplete coverage in terms of particle number, elemental composition, and mineralogy; it was possible to clearly detect regional differences in the mineralogy in single particles of suspended soil and ambient transported dust.

Sect. 2.1.1, 2.2.2, 2.1.3: please reorganize these sections to clarify the methodological aspects, while the aspects mostly linked to discussion of data should be moved to Sect.3 ("Results")
We have reorganized much of the method section. However, we retain the crystal structure and sub-composition of pure minerals as our method (rather than results) despite the fact there is some discussion of the methodology. This is because we want to make a clear separation of the methodological and scientific aspects of the measurements.

Sect. 2.2: which is the impact of the differences in the sampled size distribution compared to LAAPTOF?
We have added this statement to the start of results section 3.1:
The resulting particle concentration and size distribution is dynamic but is typically on the order of $1000cm^{-3}$ with a particle size mode at $200nm$ (See supplement S2), which is below the lower size cut of the LAAPTOF, but not the filter collection. Although the two measurement techniques are not performed on the exact same particle sizes, both measurements represent the fine fraction $(<2.5\mu m)$ of the samples, due to the size distribution of the dispersed dust.
Sect. 2.4: the vicinity of the measurement site to the airport runway has an effect on measurements?
The airport was down wind of the sampling. There were only a handful of flights per day, and while some carbonaceous aerosols were detected in association with aircraft activities, they had no impact of the mineral dust in the measurement.

Page 10, line 8: "to analyse"
We have changed this sentence.

Results and Discussion, sections 3 and 4: I invite the authors to consider if some of the aspects in Sections 2.1 should not be moved here and also if the whole presentation of results and the discussion should not be reduced a little bit in length and also made more readable for a non-advanced reader. The text is fact very dense and complicated in some points in my opinion and it could be probably simplified.

Thank you for your advice. We have combined the results and discussion sections in order to reduce the complexity and length of the text. We now have a longer, but lighter conclusion section that is more accessible to the non-advanced reader, whilst maintaining the detail in the results/discussion.

---

## Author Comment (AC2) · 23 Jan 2019

**Detailed Response to Anonymous Referee #1**
Reviewer's comments in black
Authors' response in blue
Changes in the manuscript in green

The analysis presented here could be more fairly put in context of SP-MS analysis focused on mineral dust that has come before, while still highlighting the advance made here regarding between distinguishing between mineral phases.

The manuscript now references much of the previous SP-MS analysis that you have brought to our attention. Thank you.

I think the importance of single-particle measurements of mineral phases should be developed more in the introduction to better motivate the work. This is a major challenge that limits our understanding not just of sources and transport of mineral dust, but also of their critical properties. Knowledge of mineralogy (as opposed to just elemental composition) is necessary to understand chemical reactivity, and is crucial to predict the mineral particles' ice nucleation properties.

Because this paper is likely to be of interest to a wide audience, we did not want to put too much emphasis on the ice nucleating properties, which will be further developed in a second manuscript (under preparation by R.Ullrich). However, in order to get a better balance between the dust cycle aspect and the critical properties, we have now refer to laboratory measurements, and ice nucleation more specifically in the abstract. We have also moved the introduction to the role of mineralogy in ice nucleation further up in the introduction section.

The manuscript was often hard to follow, sometimes written more like a lab report with concepts and terms suddenly introduced with no explanation or definition. It was especially difficult to keep track of where the various geographical locations that are mentioned repeatedly actually are. Many of the figures are not designed that well, do not have captions that adequately explain the figure, and could be improved in their clarity. Several sections often start with just one sentence before the subsection starts, an odd way to start a section. Many typos and syntax errors abound, and many references cite ACPD versions of papers instead of the ACP version; the manuscript needs to be carefully proofread and improved. The analysis and results presented here will certainly be of interest to the ACP community. This manuscript should be acceptable for publication in ACP once the authors fully address the questions raised and improve the manuscript's clarity

We have made a considerable effort to make the manuscript easier to follow. The results and discussion sections have been combined, the section headers are clearer and the figure captions contain more details.

Introduction: (Jickells et al., 2005) is an excellent review of the importance of atmospheric mineral dust for the oceans and biogeochem.

This reference has been added.

Pg 3/line 14: It is not really accurate to refer to the "IN fraction". A fraction of particles can be IN active at a /specific/ temperature. Most mineral phases are IN active at some mixed-phase cloud temperature, even weak ice nucleants such as quartz. Perhaps rephrase to refer to the ice nucleation properties of mineral dust particles (which requires knowledge of the specific mineral phases present).

This sentence has been re-written to: "However, relating IN properties to mineral phase in natural dust particles is much more difficult due to complex mineralogy and mixing state that is difficult to resolve."

Pg 3/line 16: When discussing ice nucleation it is important to refer to the ice nucleation mode being referred to. While strong acids and SOA added to dust particles can impair deposition freezing, they do not seem to interfere with immersion freezing. Presumably the condensate or reaction product dissolves off in the droplet (Niedermeier et al., 2011; Reitz et al., 2011; Sullivan et al., 2010a, 2010b)

The sentence was incorrect as originally written, but to avoid a detailed review of ice nucleation, the text has been re-written to refer to ice forming mechanism generally: "To complicate things further, cloud chamber studies of silicate mineral dust coated with secondary sulphate and organics have demonstrated that this mixing can alter hygroscopicity and change the ice nucleation efficiency of a particle but is dependent on the ice forming mechanism \citep{Mohler2008,Sullivan2010,Sullivan2010a,Niedermeier2011,Reitz2011}."

3/22-29: This paragraph rather discounts the many important observations that have been made regarding individual mineral dust composition, aging, and reactivity achieved through offline electron/x-ray microscopy, for example: (Hwang and Ro, 2006; Jeong and Chun, 2006; Krueger et al., 2003, 2004; Ro et al., 2005; Sobanska et al., 2012; Tobo et al., 2009; Zhang et al., 2003).

The authors agree that it this paragraph lacks appropriate summary of previous work. We have altered the text to;  "The application of these techniques can differentiate silicate and calcium rich particles and show evidence of heterogeneous reactions in the atmosphere {Ro2005,Jeong2006,Sobanska2012}. However, further differentiation of silicate mineral phase is hampered by the difficulty in leveraging the full quantitative capability of SEM due to particle morphology effects. Consequently, silicate particles are reported in compositional groups, such as the frequently used scheme described by Kandler{2009}, which describes the dominant elemental features but not actual mineral phase."

Hwang and Ro, 2006; Krueger et a.l, 2004; Sobanska et al., 2012 are specific to the heterogeneous reaction of calcium containing particles with atmospheric gasses, consequently we have included them in the section 3.2.1 which reports calcium rich particles and reaction products.

4/4: "SPMS...but not differentiate mineralogy." This is an entirely inaccurate statement. Indeed later in the paper prior use of SP-MS to differentiate mineralogy is presented. But even there much of the closely relevant prior studies using SPMS to analyze dust mineralogy is not discussed or cited. The first detailed look at how dust mineralogy influences the chemistry of atmospheric dust, as well as particle size and mixing state analysis, was presented by (Sullivan et al., 2007a) in ACP. That analysis found that Ca-rich particles accumulated more nitrate and chloride, while Fe/Al-rich particles took up more sulfate, for example. The Sullivan et al. paper cited here focuses on chlorine chemistry in mineral dust (Sullivan et al., 2007b), which is certainly relevant, but the analysis presented in the ACP paper came first and is more closely relevant to the analysis presented here. There is another paper that focuses on organic acids in mineral dust particles using SP-MS (Sullivan and Prather, 2007). I mention this series of papers that use the ATOFMS as I believe they were the first to really analyze in detail the mixing state and mineralogy of individual dust particles using SP-MS.

What was meant to be conveyed is the actual mineral phase in dust particles is not differentiated. The introduction has been modified to convey the analytical challenge associated with identifying mineral phase. This statement has been clarified to better express this:

"Despite these limitations, SPMS can characterise a particle population by classifying particle types and measuring temporal trends in particle number concentrations using cluster analysis \citep{Hinz2006, Rebotier2007,Gross2010}. Although the reported number concentration are also not fully quantitative \citep{Murphy2007}, a relative trend in certain particle types can be achieved. This techniques have been used to discriminate mineral dust particles from other refractory aerosol types such as sea salt \citep{Sullivan2007c,DallOsto2010,Fitzgerald2015,Schmidt2016}, but cannot differentiate the actual mineral phase of silicates within dust particles. More recently, a machine learning technique has shown promise with the classification of mineral dusts of similar composition \citep{Christopoulous2018}, but this techniques also requires suitable mineral dust proxies for training data.

The mixing state of dust particle is discussed in detail in section 2.1.3.

Sect. 2.1: More details regarding the particle detection system should be provided. Is this a custom non-standard setup for the LAAPTOF? Nowhere is the typical particle detected fraction (as a function of particle size) presented. This is critical information as the sub-population of total ambient particles actually detected by the instrument can significantly bias the measurements and analysis. The particle detection rate also governs the ability to observe changes in particle composition and sources over short timescales, something that is focused on here.

More details of the typical particle detected fraction are now given the methods section 2.1.

Laboratory evaluation of the fiber-coupled laser system indicate that the detection efficiency peaks at 0.25 with spherical particles \cite{Marsden2016a}, but the overall efficiency of the instrument also depends on ablation efficiency with respect to particle composition. In a study of nominally pure mineral samples, \cite{Marsden2017} reported the number of optically detected particles that produced a mass spectra (i.e. ablation efficiency or hit rate) of 0.29 and 0.14 for illite and kaolinite respectively, but was also dependent on the amount of impurities such as Titanium. Furthermore, from the authors own experience, it likely that pure quartz may have an ablation efficiency close to zero and is not considered in our analysis, but is unlikely to be a major component in the fine fraction in

any case. The potential maximum overall efficiency of the LAAPTOF measurement of clay mineral ranges from 0.0725 for pure spherical particles particles of illite, to 0.035 for pure spherical particles of kaolinite. The exact efficiency of the instrument is not known in most situations because the size, shape and composition of the particles would have to be known a priori for accurate calibration.

The relevant parameter for LDI is the laser energy power density (W/cm2). What is the laser diameter when it intersects the particle, so this quantity can be reported?

Unfortunately, the exact value if this parameter is not known to the authors. It requires technical information that has not been released by the OEM.

5/5: Just referring the reader to another paper for relevant information is not satisfactory. Please provide a clear summary here regarding the relevant performance characteristics of this LAAPTOF configuration that influences what fraction of particles are actually detected, put in the context of the aerosol populations analyzed here and their associated properties that govern particle sample and detection.

We have added a paragraph that summarises performance characteristics previously reported with this instrument with mineral dust:

"Laboratory evaluation of the fiber-coupled laser system indicate that the detection efficiency peaks at 0.25 with spherical particles {Marsden2016a}, but the overall efficiency of the instrument also depends on ablation efficiency with respect to particle composition. In a study of nominally pure mineral samples, {Marsden2017} reported the number of optically detected particles that produced a mass spectra (i.e. hit rate or ablation efficiency) of 0.29 and 0.14 for illite and kaolinite respectively, but was also dependent on the amount of impurities such as Titanium. Furthermore, from the authors own experience, it likely that pure quartz may have an ablation efficiency close to zero, so that the potential range of overall efficiency ranges from 0.0725 for spherical particles of illite, to almost zero for pure quartz particles. The exact efficiency of the instrument is not known in most situations because the size, shape and composition of the particles would have to be known a priori."

A discussion of the impact of these efficiencies on the current measurement are discussed in….

5/6: The LAAPTOF is quite similar in design to prior SP-MS instruments. I do not see why the authors think they can only refer to the few prior LAAPTOF papers that have demonstrated quantitative particle analysis. There is a large body of SP-MS work demonstrating the semi-quantitative capabilities, and even truly quantitative analysis if it is done carefully with calibration (Bhave et al., 2002; Fergenson et al., 2001; Gross et al., 2000). The use of SP-MS to determine heterogeneous kinetics is perhaps the best example of quantitative analysis (Saul et al., 2006; Sullivan et al., 2009).

This section is specifically about the LAAPTOF. We have now formed a better overview of the quantitative capabilities in the introduction.

6/1: It would be useful to expand on this weathering of minerals and how that relates to this analysis. Later in the paper feldspars are discussed as indicating less aged/weathered mineral particles, and I think that idea comes from the information presented here, but the connection is not clear. Feldspars can also be converted to the amorphous clays by acid attack, right? (Wex et al., 2014)

This was referring to the consequence of weathering within the soil on mineral composition and is therefore more relevant to the discussion of the results of the soil dust analysis.

6/15: It is not clear if this analysis works for all silicates such as quartz, which is a very

This has now been included in section 2.1 (see above).

6/23: The poor separation of K+ and Ca+ is an important limitation in this analysis. Please explain the causes of this. Is it specific to the LAAPTOF's configuration? While space charge effects that degrade ion resolution are common in LDI-MS of ion-rich mineral particles, usually K and C can be resolved. The low ionization energy of K can also produce an overly broad ion peak that extends past m/z 39.5 and into Ca at m/z 40. Is this the issue here? Is the LAAPTOF's MS too poor in resolution to resolve K and Ca?

The poor resolution results from both energy focussing and low ionisation energy, but it is not known if this typical of this instrument design. The sentence has been rephrased to:
"In addition, $Ca^{+}$ is not considered because it cannot be reliably resolved from potassium signal at m/z 39 due to peak broadening,"

6/31: The analysis discussed here is of a semi-quantitative (relative) nature, so I don't see why the authors refer to it as "non quantitative". That has a quite different meaning.

The data in figure 3 shows that the elemental composition is not even relative (feldspar appears less K rich than illite). We have clarified the paragraph to be more explicit about what is non-quantitative and what is semi-quantitative (see also comments on non-quantitative below):

Although the elemental sub-composition measurement is clearly non-quantitative with respect to bulk XRF analysis, the measurement is semi-quantitative (relative) with respect to samples of minerals with the same crystal structure. For example, a clear separation between K and Na rich feldspar is apparent in Fig \ref{tern_cal}, which is relative to their actual elemental ratios. This is not true if comparing the clay mineral illite with the framework silicate K-feldspar, which would not be easily distinguishable from each other if plotted in the same space, despite the clear differences in elemental composition. It is therefore necessary to apply crystal structure analysis to achieve semi-quantitative composition and distinguish clay minerals from feldspar when analysing natural soils with SPMS.

Sect. 2.1.2: At the end there really needs to be a summary of what mineral phases can be included in this type of analysis, and which can be distinguished. I was pretty confused as to what mineral phases (such as quartz, or carbonates) are and are not included in the analysis performed here. A good discussion of how reliably the various mineral phases (that can be analyzed) can be discriminated from each other is also

needed. Please be as quantitative there as possible

This section has been re-organised to make it clearer what mineral phases are included. The first paragraph now reads:

"Sub-compositional analysis is used to produce relative composition measurements that can be compared to fingerprints generated from nominally pure mineral samples. Here, the mineralogical composition of dust is considered with the ternary system $Al^{+}+Si^{+}, K^{+}, Na^{+}$, cations readily observed in the SPMS of mineral dust (m/z 27, 28, 39, 23 respectively), using the assumption that the fine fraction ($<2.5\mu m$) is primarily composed of aluminosilicate clays and feldspars, which is a reasonable assumption for dust derived from a continental land mass. Quartz and carbonate minerals are not considered with sub-compositional analysis due to the inability to efficiently ablate pure quartz and the apparent lack of a clear carbonate signal respectively. In ambient dust, calcium rich particles are considered separately to aluminosilicate particles."

We have also clarified the approach in this paragraph:

Although the elemental sub-composition measurement is clearly non-quantitative with respect to bulk XRF analysis, the measurement is semi-quantitative (relative) with respect to samples of minerals with the same crystal structure. For example, a clear separation between K and Na rich feldspar is apparent in Fig \ref{tern_cal}, which is relative to their actual elemental ratios. This is not true if comparing the clay mineral illite with the framework silicate K-feldspar, which would not be easily distinguishable from each other if plotted in the same space, despite the clear differences in elemental composition. It is therefore necessary to apply crystal structure analysis to achieve semi-quantitative composition and distinguish clay minerals from feldspar when analysing natural soils with SPMS.

A discussion of the reliability of the discrimination of mineral phases has been included in the discussion.

7/7: Reactions with ozone will just convert O3 to O2 on dust, and not add any material.

The reference to ozone in this sentence has been removed:
"Internal mixing of non-mineral species can occur during soil formation or during transport in the atmosphere where heterogeneous reactions take place on the surface of the particle {Usher2003}."

7/10: Usually see other more specific ion markers for organics in dust using SP-MS. m/z +43 is common for oxidized organics, and negative ions often have fragments from organic acids (Silva and Prather, 2000; Sullivan and Prather, 2007). Please discuss why only very small organic fragments are observed here. The LDI laser pulse energy of 3-5 mJ is rather high and perhaps caused extensive fragmentation. Also, while organics can "char" to EC ions, C2+ could also come from black carbon that was mixed with the particle. I would be wary of using C2+ to identify organic carbon, it is a generic carbonaceous (OC + BC) marker.

A sentence that refers to extensive fragmentation has been added to the paragraph:
"Only small organic fragments are observed due to the extensive fragmentation of organic molecules."

We do not actually use the C2 marker to identify organics, we are just pointing out which ion combinations appear in the spectra after the addition of organic material.

7/19: More unique biological ion markers are often observed, such as phosphates, as well as the rather ubiquitous (Murphy et al., 2006) CN- and CNO- ions, if you look at the spectra presented from ATOFMS analysis for example (Creamean et al., 2013; Pratt et al., 2009; Sultana et al., 2017).

Phosphate markers are not unique to biological material as they can be derived from in-organic minerals such as apatite. The following line has been added to the text to clarify: "Phosphate marker $PO_{3}^{-}$ is not considered because it could be derived from the calcium phosphate mineral apatite as well as biological material."

7/23: The use of chloride in the analysis is confusing and needs more discussion. Chloride is not only a part of primary mineral components but can also accumulate in dust via transport and uptake of HCl(g) etc. Please better explain the purpose of using Cl- in the analysis, and how that would be affected by secondary sources of Cl in the dust particles.

The sentence has been re-written to:
"The Cl- elemental ion is included despite it also having mixed provenance such as primary chlorides or secondary uptake of $HCl$, because it is preferentially ionised due to very high electron affinity and therefore is included as a reference that would otherwise perturb the measurement. It is also an indication of the purity of silicate particles is as pure fresh silicate should not contain Chlorine."

8/14: Rationing the signals to Si seems to follow the work of Sullivan et al. that normalized to Al. The use of Si is likely more appropriate, especially in this work's focus on silicate minerals. There is also other work that introduced ternary analysis to understand mineral dust composition and aging (Krueger et al., 2003, 2004; Laskin et al.,2005; Yuan et al., 2004).
In this particular case, the text is referring to quaternary analysis with the SEM technique.

Sect. 2.3: Odd to just have one sentence here. It also inaccurately implies that distinguishing between different minerals using SP-MS has never been done before.

This sentence has been removed.

9/14: what are these, and how to they relate to mineral composition? "fluvisols (50%), associated with yermosols (20%), regosols (20%) and solonchaks (10%)."

To avoid a lengthy explanation, the reference to soil types has been removed.

9/15: It took me awhile to realize that these are the locations that each sample was taken from. Would probably be better in a Table.

This has been put into a table.

11/3: greater sensitivity to alkali metals is due to low ionization energy? Please explain. There is too much expected expert knowledge that non SP-MS users will not necessarily know.

This sentence now refers to ionisation energy:
"…a much greater sensitivity to alkali metals in the SPMS measurement than in the established filter technique due to low ionisation energy."

11/7: where were these soil samples from? Also North Africa?

Yes North Africa, this has been clarified in the text.

11/12: felsic means felspars?

Feldspar like in composition.

11/19: This is a good example of where a brief reminder of what this "Anti-Atlas" location is would be useful.

Changed to: "Anti-Altas mountain range."

12/13: This is why a proper discussion of the LAAPTOF's detection efficiency versus particle size (for dust particles) is needed. This suggests that the instrument is only detecting 1% of total silicate particles. How much of this is just due to particle size transmission issues, versus the instrument's actually particle hit percentage?

The detection efficiency of the system is now discussed in greater detail in section 2.1.

12/23: Uptake of HCl by dust observed using SP-MS was a focus of (Sullivan et al., 2007b), which you cite yet oddly do not discuss when very relevant here.

13/2: Tenerife?
"the island of Tenerife in the North Altantic"

13/31: What metric is "> 0.2"? Ion peak area? How do you decide if a particle has a "significant fraction (> 0.2)"?

The mixing state sub-composition is defined by the ternary system Chlorine - Org-bio – Nitrate as defined in the paragraph above. Org-Bio > 0.2 simply refers to the number of particles where the org-bio fraction in this sub composition is greater than 0.2 (or 20%). The sentence has be re-worded to:

"The number of particles whose mixing state sub-composition contained more than 20% organic-biological material (Org-Bio > 0.2) varies with the dust concentration"

14/11: This dust mobilization refers to the emissions of dust or its transport? Confusing.

Changed to: "dust emission"

14/23: Can also have nitrate from coagulation with ammonium nitrate, uptake of N2O5,etc. (Korhonen et al., 2003; Sullivan et al., 2007a; Zhang et al., 2000). Can't conclude it is all from HNO3.

Changed to: "The mixing of nitrate with silicate during and after D1 indicates contact with polluted air and is consistent with transport from the North"

14/25: By biogenic source are you referring to sulfate derived from DMS? Explain.

Changed to: "An increase in the fraction of sulphate containing sea-spray aerosol in D2 on the other hand (Supplement S3.5) may be associated with organosulfur containing compounds from biogenic sources in the coastal upwelling region off the coast of Mauritania."

15/11: Some key references to whole rock geochemical analysis would be nice.

Added a reference to ternary analysis in the context of whole rock geochemical analysis. (Pawlowsky-Glahn, 2006)

15/13: Important paper on ternary analysis: (Yuan et al., 2004)

Thank you, this important reference is now included.

15/14: Matrix effects in LDI are mentioned repeatedly but never explained or discussed. Non-experts will be unfamiliar with this important effect.

"The matrix effects arise from the incomplete desorption and ionisation process and the competitive ionisation of atoms and molecular fragments, so that co-variance of analyte signals is relative to ionisation energy and electron affinity of the surrounding matrix \citep{Reinard2008}. In circumstances where the composition of the matrix is known a priori, careful calibration with a suitable proxy can produce quantitative or semi-quantitative measurements of an analyte within a single particle \citep{Gross2000,Bhave2002}. However, if the particle matrix is complex, such as in soils and tranported dust, a this type of calibration cannot be made due to the lack of a suitable proxy."

15/28: Is the reader supposed to know where Praia Cabo Verde is? If so I have forgotten so a reminder would be useful

Praia, Cabo Verde is described as the location of our measurements in section 2.4

Page 15: Fractionation of mineralogy versus particle size during transport is never discussed (Arimoto et al., 2004; Gong et al., 2003; Mori et al., 2003). Nor is the size distribution of the different mineral dust types presented, which is a real oversight.

The properties of the dispersed dust is now discussed in the Results and Discussion (Section 3.1), with example given in the supplement.

The fractionation of mineralogy vs particle size is included in the introduction where we make a distinction between the fine and coarse fractions. We now state at the start of the methods section the assumption that

"the fine fraction (<2.5\mu m) continental sediment is primarily composed of aluminosilicate clays and feldspars"

15/34: Quantitative information regarding by how much the mineral composition can change over just one hour would be very useful here. How significant a change can occur?

In most of our ambient measurements, relatively large numbers of illite rich particle matrix (ISCM Ratio > 5) suggests a dust source on the NW margins of the Sahara during the summer. However, a rapid change (< 1 hour) towards a felsic/amorphous particle matrix (ISCM Ratio < 1) was observed when back-trajectories suggest direct emission into the marine boundary layer from the West African coast.

16/20: What are these accompanying aircraft sorties? Another example of new ideas that are just sort of thrown out there without proper introduction or explanation.

Changed to:

but it is interesting to note that Price 2018 did not see significant variation in ice nucleating particle (INP) concentration in aircraft based studies of the Saharan air layer during ICE-D, despite geographically widespread sources of that dust.

16/22: As mentioned above, there are more and more specific biological ion markers to use than just CN- and CNO-. Also please summarize the analysis method developed by Zawadowicz, so it can be better understood why it might not be transferrable from the PALMS to LAAPTOF. If both use the same LDI laser wavelength it is likely transferrable.

As discussed earlier, the phosphate ions are not specific to biological material. We did not have the resource to fully test the method of Zawadowicz.

Conclusions: Mentioning the timescale and magnitude under which changes in mineralogy occur would be good information to include here. That seems to be one of the major findings from the ambient measurements.

Due to the re-organisation of the discussion into the methods section, the conclusion is now longer. It now includes this paragraph:

In most of our ambient measurements, relatively large numbers of illite rich particle matrix (ISCM Ratio > 5) suggests a dust source on the NW margins of the Sahara during the summer. However, a rapid change (< 1 hour) towards a felsic/amorphous particle matrix (ISCM Ratio < 1) was observed when back-trajectories suggest direct emission into the marine boundary layer from the West African coast. This episode lasted only a few hours and challenges previous findings from off-line measurements that the source and composition of transported dust only changes on a seasonal basis.

17/9: Really semi-quantitative, not non quantitative. Almost all the analysis presented here is quantitative-based, not just qualitative.

In the extended conclusion section, we have paid greater attention to the quantitation issue in this new paragraph:

These measurements were made under the reasonable assumption that single particles in the fine fraction were composed of either clay minerals or feldspars/amorphous matrix, a distinction that can be realised by the novel crystal analysis technique. Although the SPMS technique is shown to be generally non-quantitative with respect to the elemental sub-composition of pure mineral phases such as illite and K-feldspar, a semi-quantitative (relative) measurement of elemental composition can be achieved after particles are separated into mineral groups based on crystal structure. Further differentiation of mineral phase can then be made by comparison to pure mineral fingerprints from within the mineral group. This indicates the importance of particle structure in addition to particle composition in the matrix effect in SPMS.

17/14: Understanding individual dust mineralogy is also important for understanding reactivity!

Changed the following sentence to:

These example spectra of transported dust should also be useful for studies of ice nucleation, radiative properties, and in-homogeneous processes of dust,

Fig. 1: Explaining the color code for the ions in the caption would be useful.

Color code now included.

Fig.2: Should cite your prior paper in the caption where this method was developed so the connection is clear.

Citation now included.

Fig. 4: The phosphate marker I mentioned above for biological is evident here, why is it not also used? Granted there are mineral sources of phosphate, but it could be used in combination with CN- and CNO-.

See discussion on in-organic phosphate above.

Fig. 6: I could not find the point for pure quartz. Explain in caption that big symbols are for reference samples, and make the symbol line thicker so they are easier to see. Also add the sample name to the top of each plot as in Fig. 7.

This figure has been updated as suggested.

Fig. 7: Please explain what this means: "The color function is proportional to the Tau parameter of crystal structure which is also displayed as a histogram"

The caption has been updated:
The color function is proportional to the $\tau$ parameter as defined by the crystal structure analysis technique. The distribution of $\tau$ for each sample is also displayed as a histogram (d, h).

Fig. 10: Hard to see symbols. Make lines thicker and use a different color.

The symbols have been changed.

References: Many cite the ACPD version instead of the ACP one. Please correct.

Corrected

Pg 2/line 21: "affects" not effects

Corrected

---

## Author Comment (AC3) · 23 Jan 2019

**Detailed Response to Anonymous Referee #2**

Reviewer's comments in black
Authors' response in blue
Changes to the manuscript in green

Major comments: I would like to see more discussion of uncertainties in these measurements. In particular, the manuscript mentions that the LAAPTOF instrument tends to undercount silicate particles (page 12, line 13). Is there any evidence that it could undercount selectively and thus introduce a bias into the results as presented? In other words, are there any chemical biases in the way LAAPTOF detects silicate-rich particles?

There is certainly evidence that the instrument undercounts selectively, and We do have a sentence in the results section of the ambient measurement that alludes points out the temporal evolution is more important than the number counts.

Note that these fractions are relative to the detection efficiency of the instrument to each particle type, but the temporal evolution is representative.

However, we agree that it is not explicit what impact this has on the data. We have added the following paragraph to the Methods Section 2.1 to summaries what we know about the selectivity of the techniques regarding mineral types:

Laboratory evaluation of the fiber-coupled laser system indicate that the detection efficiency peaks at 0.25 with spherical particles \cite{Marsden2016a}, but the overall efficiency of the instrument also depends on ablation efficiency with respect to particle composition. In a study of nominally pure mineral samples, \cite{Marsden2017} reported the number of optically detected particles that produced a mass spectra (i.e. ablation efficiency or hit rate) of 0.29 and 0.14 for illite and kaolinite respectively, but was also dependent on the amount of impurities such as Titanium. Furthermore, from the authors own experience, it likely that pure quartz may have an ablation efficiency close to zero and is not considered in our analysis, but is unlikely to be a major component in the fine fraction in any case. The potential maximum overall efficiency of the LAAPTOF measurement of clay mineral ranges from 0.0725 for pure spherical particles particles of illite, to 0.035 for pure spherical particles of kaolinite. The exact efficiency of the instrument is not known in most situations because the size, shape and composition of the particles would have to be known a priori for accurate calibration.

We also now make it clear in the first paragraph of the Conclusions that the measurements do not provide complete quantitative coverage of the mineralogy of all mineral dust:

Despite the fact that the technique provides incomplete coverage in terms of particle number, elemental composition, and mineralogy; it was possible to clearly detect regional differences in the mineralogy in single particles of suspended soil and ambient transported dust.

And later in the conclusion we add the following for emphasis:

Although semi-quantitative in terms of particle number fractions due to number counting bias effects associated with instrument function, the relative temporal trends are very informative.

In the analysis of dust mixing state, chlorine, CN- and CNO- (termed "org-bio") and sulfate (later nitrate) were chosen as mixing state markers. Comparing between soils collected from the ground and particles analyzed in-situ for these particular components is complicated because of atmospheric processing, but the text seems to draw an equivalence here. For example, CN- and CNO- might indicate a biological or biogenic fraction for soils, but in situ they are much more likely to arise during atmosphericprocessing and using them as biological markers leads to large overestimates. Similarly, the large chlorine fractions at Cabo Verde are largely expected because of marine influence at that sampling location, but their origin is likely very different in the laboratory soils collected in-land.

We totally agree. It was not our intention to draw equivalence, but to point out that mineral dust particles are already mixed before emission and transport. The text failed to do that explicitly. We have added the following to the Results/Discussion
*Section 3.3 Temporal evolution of the mixing state of silicate particles (ICE-D)*
The mixing of silicate and non-silicate within single particles may result from processes within the native soil (primary), or during atmospheric transport (secondary). The analysis of the suspended soil dust (INUIT09) shows mineral particles in North African soils are already mixed, particularly with varying quantities of chlorine, sulphate and organic/biological material.

We think the nitrate mixing has a large influence from the atmosphere due to the low level of variation in the soil compared to the ambient measurements. We have added to the conclusions:
Internal mixing state was of some use to understanding transport history of ambient dust, but must be used with caution because some degree of mixing was already present in the primary soil. However, variations in internally mixed nitrate suggested dust from the NW margins of the Sahara was deposited into the marine boundary layer after transport in the Saharan air layer.

Minor comments: In section 2.1.1, large parts of the text (especially first two paragraphs read like introductory material instead of methods.

Agreed. We have moved the description of the crystal structure of the common minerals to the introduction.

Page 8, line 17: Figure 6 is called out in the text before Figures 4 or 5.

This has been corrected.

General comment for the methods section: please indicate the number of single particles analyzed in laboratory and field studies.

The number of particles has been added to the captions where appropriate.

Results, first sentence: "...we choose to analysis...
" should be we chose to analyze? Please clarify.
The sentence no longer contains this sentence as it is already covered in the Methods section.

Section 3.1, line 12: the authors say that vast majority of particles contained silicate markers, but then they also say that all particles contained some silicate minerals. This is a bit vague.
This sentence has been removed. The first paragraph of the Results section now gives an overview of the particle size distribution of the suspended dust, and the coverage of the two techniques. This was in response to comments from the other reviewers.

Page 11, lines 2-5: I am not sure I follow the reasoning here and in Figure 8. For the Moroccan sample, the two techniques seem to be showing the exact opposite composition.
This was not well explained in the body text or the caption. We have re-written the body text to the following:

A comparison of the sub-composition (Al+Si)/(Al+Si+K+Na) obtained by SPMS and SEM measurement (Fig. \ref{SEM_SPMS}) demonstrates a much greater sensitivity to alkali metals in the SPMS measurement (due to low ionisation energy) than in the established filter technique. The SEM techniques show a lower (Al+Si)/(Al+Si+K+Na) ratio in the Moroccan sample (DDS01, panel (a)) compared to the Sahel sample (SDN02, panel (C)), but this is greatly exaggerated in the SPMS analysis (panels (b) and (d)) due selective ionisation of K and Na and the matrix effect.

And the caption to Figure 8:

A comparison of the relative sensitivity of the SPMS and SEM techniques to the principal elements in silicate minerals. Histograms represent the sensitivity to alkali metals of the interstitial complex with respect to the Al and Si of the silicate structure ((Al+Si)/(Al+Si+K+Na) in single particles. Moroccan soil sample DDS01 (a, b) compared to Sahelian soil sample SDN02 (c, d) using the SEM and SPMS technique respectively.

Section 3.2, line 9: Why were peaks shifted in positive and not negative spectra?
Peaks shifted in both positive and negative ion modes. It was just the positive mode where the shift was greater than 0.5da, and hence affected unit mass assignements.

Page 15, line 10: "Sub-compositional analysis is a techniques" – should be technique.
Corrected!

---

## Editor Decision (ED1)

**1) Editor comment:** I can't find the following change in the revised manuscript.

Reviewer #2: Pg 3/line 14: It is not really accurate to refer to the "IN fraction". A fraction of particles can be IN active at a /specific/ temperature. Most mineral phases are IN active at some mixed-phase cloud temperature, even weak ice nucleants such as quartz. Perhaps rephrase to refer to the ice nucleation properties of mineral dust particles
(which requires knowledge of the specific mineral phases present).

This sentence has been re-written to: "However, relating IN properties to mineral phase in natural dust particles is much more difficult due to complex mineralogy and mixing state that is difficult to resolve."

**2) Editor comment:** In your response below, it seems that you intended to add a discussion of the efficiencies. Was this actually done?

Reviewer #2: 5/5: Just referring the reader to another paper for relevant information is not satisfactory. Please provide a clear summary here regarding the relevant performance characteristics of this LAAPTOF configuration that influences what fraction of particles are actually detected, put in the context of the aerosol populations analyzed here and their associated properties that govern particle sample and detection.

We have added a paragraph that summarises performance characteristics previously reported with this instrument with mineral dust:

"Laboratory evaluation of the fiber-coupled laser system indicate that the detection efficiency peaks at 0.25 with spherical particles {Marsden2016a}, but the overall efficiency of the instrument also depends on ablation efficiency with respect to particle composition. In a study of nominally pure mineral samples, {Marsden2017} reported the number of optically detected particles that produced a mass spectra (i.e. hit rate or ablation efficiency) of 0.29 and 0.14 for illite and kaolinite respectively, but was also dependent on the amount of impurities such as Titanium. Furthermore, from the authors own experience, it likely that pure quartz may have an ablation efficiency close to zero, so that the potential range of overall efficiency ranges from 0.0725 for spherical particles of

illite, to almost zero for pure quartz particles. The exact efficiency of the instrument is not known in most situations because the size, shape and composition of the particles would have to be known a priori."

A discussion of the impact of these efficiencies on the current measurement are discussed in….

**3) Editor comment:** The reviewer commented on your sentence 'Similarly, organic markers $C_2$;$C_2H$ and $C_2H_2$(m=z24;25;26) appear on particles after mixing suspended feldspar with ozone and _-pinene (Fig. 4(c)). (now p. 8, l. 13) as below. Your response does not match the reviewer's comment as your wording still implies that C2 can be used as an organic marker.

Reviewer #2, 7/10: Also, while organics can "char" to EC ions, C2+ could also come from black carbon that was mixed with the particle. I would be wary of using C2+ to identify organic carbon, it is a generic carbonaceous (OC + BC) marker.

We do not actually use the C2 marker to identify organics, we are just pointing out which ion combinations appear in the spectra after the addition of organic material.

**4) Editor comment:** The following comment has not been addressed at all. (now: P. 13, l. 25)

Reviewer #2: 12/23: Uptake of HCl by dust observed using SP-MS was a focus of (Sullivan et al., 2007b), which you cite yet oddly do not discuss when very relevant here.

**5) Editor comment, follow-up on** Reviewer #3: Abstract, line 19-20: please clarify the sentence now: p. 1, l. 22-3: This sentence still sounds strange since the subject in the two fragments changes and it is not clear what a 'continuous difference' is.
Maybe better something like:
In most cases, the differences in the mineralogical composition between particles within a soil sample were small. Thus, particles were not composed of discrete mineral phases.
* * *
**Minor editor comments**

p. 3, l. 30: consist

p. 3, l. 35: of these cations…

p. 3, l. 35: charge-balanced

p. 4, l. 3: to be abundant

p. 4, l. 7: is chemically similar…

p. 4, l. 24: either 'shows' or '…these techniques show evidence…'

p. 4, l. 28: describes → describe

p. 4, l. 29: semicolon is redundant

p. 5, l. 12: 'concentration is…' or 'concentrations are…'

p. 5, l. 12: 'achieved' does not seem to be the right word here. Replace by 'observed' or 'detected'

p. 5, l. 13, and l. 16: These techniques…

p. 6, l. 29: either 'produced a mass spectrum' or 'produced mass spectra'

p. 6, l. 30: authors' own …

p. 6, l. 31: verb missing (it is likely…(?))

p. 6, l. 33: 'particles' is redundant

p. 8, l. 27: Is it really SO4- or rather $SO_4^{2-}$?

p. 8, l. 29: it is therefore…

p. 9, l. 15: per sample

p. 9, l. 19: skew and suppress

p. 9, l. 20: remove the last 'ratios'

p. 11, l. 17: …are dynamic

p. 12, l. 16; p. 13, l. 29: Is it really $Ca^+$ or $Ca^{2+}$?

p. 12, l. 12: due to…

p. 14, l. 18: are much reduced…

p. 14, l. 23: consideration

p. 14, l. 29: which section are you referring to?

p. 17, l. 16: measurements show

p. 17, l. 27: either '…numbers suggest' or 'number suggests…'

Figure 9, caption: should this read $0.58 < \tau < 0.8$>

Figure 9: Add a) , b) … to the individual figures

Figure 12, caption: "A total of 12698 (a) concentration of silicate and calcium rich particles (total measurement of 12698 silicate and 6837 calcium rich particles)." - Please reword and clarify.

Figure 13, caption: - (1301 dust particles analysed)

                          - back trajectory analysis suggests

References : Ahern et al : Cite AMT paper instead of AMTD

---

## Author Response (AR2)

**Editor Comment 1.**

Apologies, I removed this statement when I re-organised the introduction. I now have the following sentence at the end of the third paragraph of the introduction (Page line) which provides a better flow from IN properties of dust into the introduction of dust source:

This highlights the need for measurement of single-particle composition, including both mineralogy and internal mixing state, but is extremely challenging in natural dust particles due to complex mineralogy and mixing state that is a product of the source area, emission mechanism, and atmospheric processing.

**Editor Comment 2.**

There is a note about particle number fractions at the end of the first paragraph section 3.22. (Page line) I have elaborated on this in the latest version:

Note that these fractions are relative to the detection efficiency of the instrument to each particle type (see Section 2.1), so that ratios presented here are semi-quantitative only, but the temporal evolution is representative of changes in particle composition.

Note also that the semi-quantitative nature of the of the number fractions are also pointed out in the conclusions (now page line ).

**Editor Comment 3.**

I have now added to the sentence that follows the discussion of C2 peaks at m/z 24.

These ion combinations have also been observed on ambient mineral dust using SPMS instruments {Silva2000, Sullivan2007c, Fitzgerald2015}, but may also be derived from black carbon mixed with the particle as well as charred organics.

**Editor Comment 4.**

We have now added the following comment to section 3.1.2.:

All laboratory suspended soil samples contained particles with significant amounts of chlorine, demonstrating that chlorine can be internally mixed with

silicate particles before emission and transport. This makes the Al/Cl ratio used by {Sullivan2007} with Asian dust, an unreliable measure of heterogeneous reaction of silicate with atmospheric HCl in ambient Saharan dust.

**Editor Comment 5.**

The sentence in the abstract has been changed as suggested:

In most cases, the differences in the mineralogical composition between particles within a soil sample were small. Thus, particles were not composed of discrete mineral phases.

**Minor Editor Comments:**

Typos/gramma corrected. Please note, we use the notation Ca+ because the ion is singly charged in the mass spectra, following the notation used by Murphy (2006).

**References**

Murphy, D. M., D. J. Cziczo, K. D. Froyd, P. K. Hudson, B. M. Matthew, A. M. Middlebrook, R. E. Peltier, A. Sullivan, D. S. Thomson, and R. J. Weber (2006), Single-particle mass spectrometry of tropospheric aerosol particles, J. Geophys. Res., 111, D23S32, doi:10.1029/2006JD007340.

[revised manuscript text omitted]